# Effects of different land use and land cover data on the landslide susceptibility zonation of road networks

Bruno M. Meneses[1], Susana Pereira[1], Eusébio Reis[1]

[1]Centre for Geographical Studies, Institute of Geography and Spatial Planning, Universidade de Lisboa, Edif. IGOT, Rua Branca Edmée Marques, Lisboa,1600-276, Portugal

*Correspondence to*: Bruno M. Meneses (bmeneses@campus.ul.pt)

**Abstract.** This work evaluates the influence of land use and land cover (LUC) data with different
properties on the landslide susceptibility zonation of the road network in the Zêzere watershed (Portugal). The Information Value method was used to assess the landslide susceptibility using two models: one including detailed LUC data (the Portuguese Land Cover Map - COS) and the other including more generalized LUC data (the Corine Land Cover - CLC). A set of fixed independent layers were considered as landslide predisposing factors (slope angle, slope aspect, slope curvature, slope over area ratio, soil,
and lithology), while the COS and CLC were used to find the differences in the landslide susceptibility zonation. A landslide inventory was used as a dependent layer, including 259 shallow landslides obtained from the photointerpretation of orthophotos from 2005, and further validated in three sample areas. The landslide susceptibility maps were assigned to the road network data and resulted in two landslide susceptibility road network maps. The models' performance was evaluated with prediction and success
rate curves and the area under the curve (AUC). The landslide susceptibility results obtained in the two models present a high accuracy in terms of the AUC ($> 90\%$), but the model with more detailed LUC data (COS) produces better results in the landslide susceptibility zonation on the road network with the highest landslide susceptibility.

**Keywords** LUC; LUC data properties; landslide susceptibility; road networks disruption, Information Value method.

# 1 Introduction

Landslides are natural processes that can constrain the free movement of people and goods when they directly or indirectly affect road networks (Bíl et al., 2014, 2015; Hilker et al., 2009; Winter et al., 2013). The total or partial blockages of road networks have economic and societal impacts, particularly on the direct damage to the infrastructure (material damages), on the population (injuries and deaths) when driving on the affected infrastructures (Guillard and Zêzere, 2012; Pereira et al., 2014, 2017), or by causing indirect damages, such as delays, detours, material damage, and the rising prices of raw materials (Zêzere et al., 2008; Bíl et al., 2014, 2015; Jenelius and Mattsson, 2012; Winter et al., 2016).

Landslide susceptibility assessment is crucial to identifying locations with higher probabilities of landslide occurrence (Conforti et al., 2014; Guillard and Zêzere, 2012; Guzzetti et al., 2006; Pereira et al., 2014; van Westen et al., 2008). Landslide susceptibility is the likelihood of a landslide occurring in an determined area controlled by local terrain conditions; it may also include a description of the velocity and intensity of an existing or potential landslide (Fell et al., 2008; Günther et al., 2013; Guzzetti et al., 1999). Landslide susceptibility reflects the degree to which a terrain unit can be affected by future slope movements (Günther et al., 2013).

In general, the choice of landslide predisposing factors and the main details of the geographical information are not explained in a landslide susceptibility assessment based on statistical methods; rather, criteria defined in the literature (e.g., slope angle, slope aspect, slope curvature, soil, lithology, land use and land cover) are used for this selection because it can explain the occurrence of slope movements in the study area (Blahut et al., 2010; Castella et al., 2007; Castellanos Abella, 2008; Guzzetti et al., 1999, 2006; Soeters and van Westen, 1996; van Westen et al., 2008; Zêzere et al., 2008, 2017).

Beyond the influence of different environmental factors (e.g., lithology, slope angle, slope morphology, topography, soils, and hydrology) on the spatial distribution of landslides, land use and land cover (LUC) dynamics are also an important factor on landslide susceptibility assessment (Guillard and Zêzere, 2012). Certain land use and land cover changes (LUCC) (e.g., deforestation, slope ruptures to road construction, steep slopes) increase the number of unstable slopes (Reichenbach et al., 2014), i.e., promoting the propensity for landslide occurrence, and can have an important impact on landslide activity (Beguería, 2006; Glade, 2003; Mugagga et al., 2012; Persichillo et al., 2017; van Westen et al., 2008).

The LUC, while a proxy variable, is very dynamic over time and is influenced by climate-driven changes and direct anthropogenic impacts (Promper et al., 2014). In this regard, it is an important predisposing factor to landslide susceptibility assessment, and Dymond et al. (2006) mention that importance: "the quality of the input land-cover map is important because the main purpose of the landslide susceptibility model is to identify where land cover needs to be changed."

For instance, performing a landslide susceptibility analysis with a historical inventory over long periods (e.g., decades) demands the use of a permanent set of predisposing factors along the landslide inventory timeline. LUC can change over time; due to this reason, it will be more accurate to use the LUC for different periods (Reichenbach et al., 2014) than using the most recent LUC map, to avoid spatial relations between past slope instability and incorrect LUC classes.

The scale of the predisposing factors directly influences the map elements' representation and detail, as well as the choice of the scale of analysis of the final results (Leitner, 2004; Stoter et al., 2014). The choice in the level of detail will also constrain the modeling results. For example, Meneses et al. (2018b, 2018c) obtained different LUCC results in the Portuguese territory due to the use of different LUC datasets, namely the Corine Land Cover (CLC) and the official Land Cover Map of Portugal (Portuguese designation and acronym: *Carta de Ocupação do Solo*, COS), with different properties concerning the scale (1:100 000 and 1:25 000, respectively), minimum mapping unit (25 and 1 ha, respectively), and generalization level (Table 1).

Due to the variation of the road network morphology (the length vs. width of the roads), the selection of appropriate data that integrates the analysis of road blockages caused by landslides requires a systematic assessment of the detailed properties of the landslide predisposing factors (Drobnjak et al., 2016; Imprialou and Quddus, 2017; Kazemi and Lim, 2005; Orongo, 2011) to obtain detailed landslide susceptibility results at the local scale (roads).

In this context, the main goal of this work is to evaluate the influence of the LUC data properties on the landslide susceptibility zonation of road networks. Two specific goals were defined: (i) to evaluate and quantify the landslide susceptibility results using two LUC datasets (CLC 2006 and COS 2007) with different properties (scale and minimum mapping unit) in two landslide susceptibility models; (ii) to use

the output results of the two landslide susceptibility models to identify the sections of the main road network with the highest landslide susceptibility that will suffer future road blockages.

## 2 Material and methods

### 2.1 Study area

This study was performed in the Zêzere watershed (5 063.9 km$^2$) located in the Center region of mainland Portugal (Fig. 1). The north-northwest sector of this watershed is occupied by the Estrela Mountain, reaching a maximum elevation of 1 993 m, where steep slopes can be found; in the Central sector, the relief is less irregular when compared to the previous sector, but it still has steep slope areas (e.g., the vicinity of the Castelo de Bode and Cabril reservoirs); in the south-southwest sector, gentle slopes and
flat areas are predominant.

The soils of the Zêzere watershed are very variable between the north-northwest, center, and southwest sectors. In the northwest sector, cambisols predominate, with small areas of fluvisols and eutric lithosol along the Zêzere river. In the central area, lithosols are dominant, with some areas of cambisols. In the south-southwest sector, there are areas of lithosols intercalated with cambisols and luvisols.

According to the CLC 2006, the predominant LUC in the study area are forest and seminatural areas, which represent 72 % of the watershed area. Other LUC types are less representative, for example, agricultural land (25.5 %), artificialized land/urban areas (1.5 %), and water bodies (1 %), including an important fresh water reservoir, the Castelo de Bode dam (Meneses et al., 2015a). The LUC of this watershed is very dynamic, highlighting the LUCC in forest and agricultural areas derived from multiple
socioeconomic driving forces (Meneses et al., 2017) and the degradation of vast forest areas by wildfires (Meneses et al., 2018a).

Due to the large extension of this watershed, three sample areas were selected according to the high density of landslides observed in these locations: the Estrela Mountain, Vila de Rei, and Ferreira do Zêzere municipalities (areas of 86.7, 191.5, and 190.4 km$^2$, respectively), where fieldwork was developed
to validate part of the landslide inventory and the disruption of roads caused by landslides.

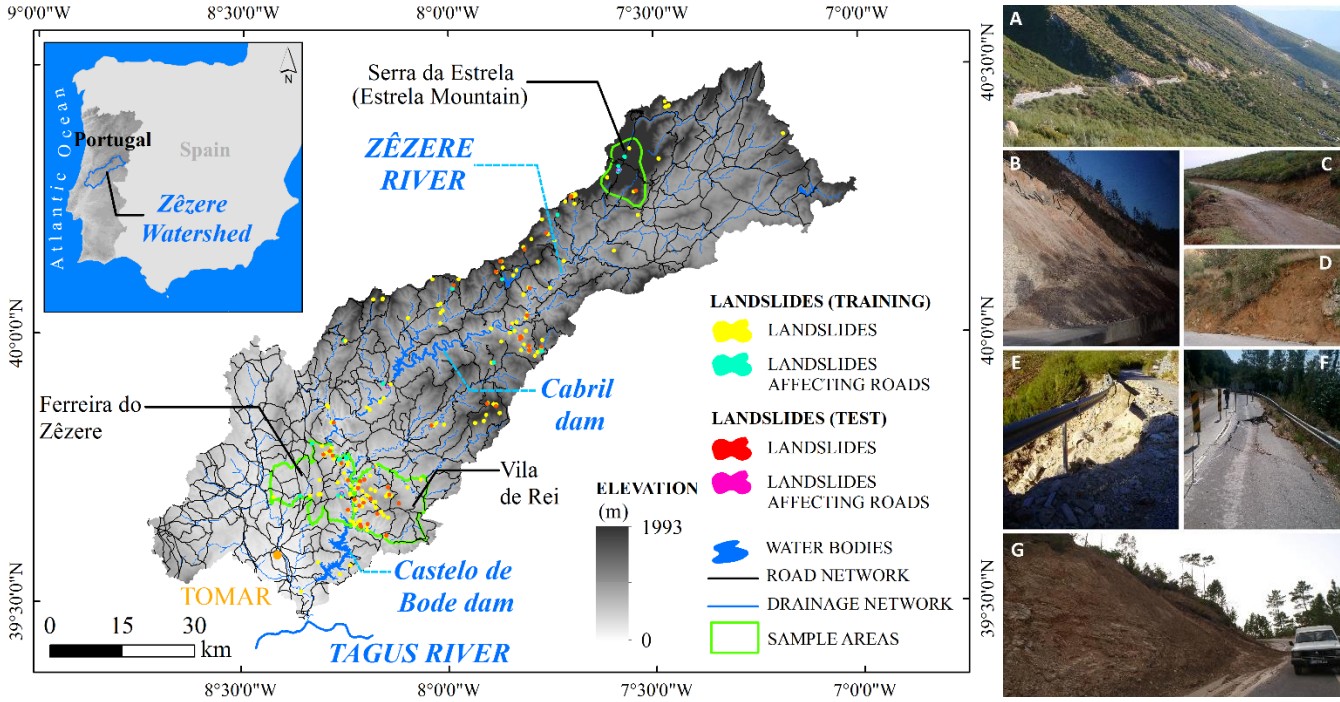

**Figure 1.** Zêzere watershed and landslide inventory. The pictures represent landslides that affected roads: A, B, C, D and E - municipality roads of Estrela mountain; F – Ferreira do Zêzere; G – Vila de Rei.

## 2.2 Data

The landslide predisposing factors used to model the landslide susceptibility in the Zêzere watershed were selected after reviewing the literature about the causal factors of landslides occurrence (Blahut et al., 2010; Castella et al., 2007; Castellanos Abella, 2008; Guzzetti et al., 1999; Reichenbach et al., 2018; Soeters and van Westen, 1996; van Westen et al., 2008; Zêzere et al., 2008, 2017) (Figure 2).

Six fixed landslide predisposing factors were considered: slope angle, slope aspect, slope curvature, slope over area ratio (SOAR), soil, and lithology. The LUC of the COS and CLC were used to find the differences in the landslide susceptibility zonation. The set of landslides predisposing factors and the corresponding classes (Fig. 2) were the same in all models, only changing the LUC data.

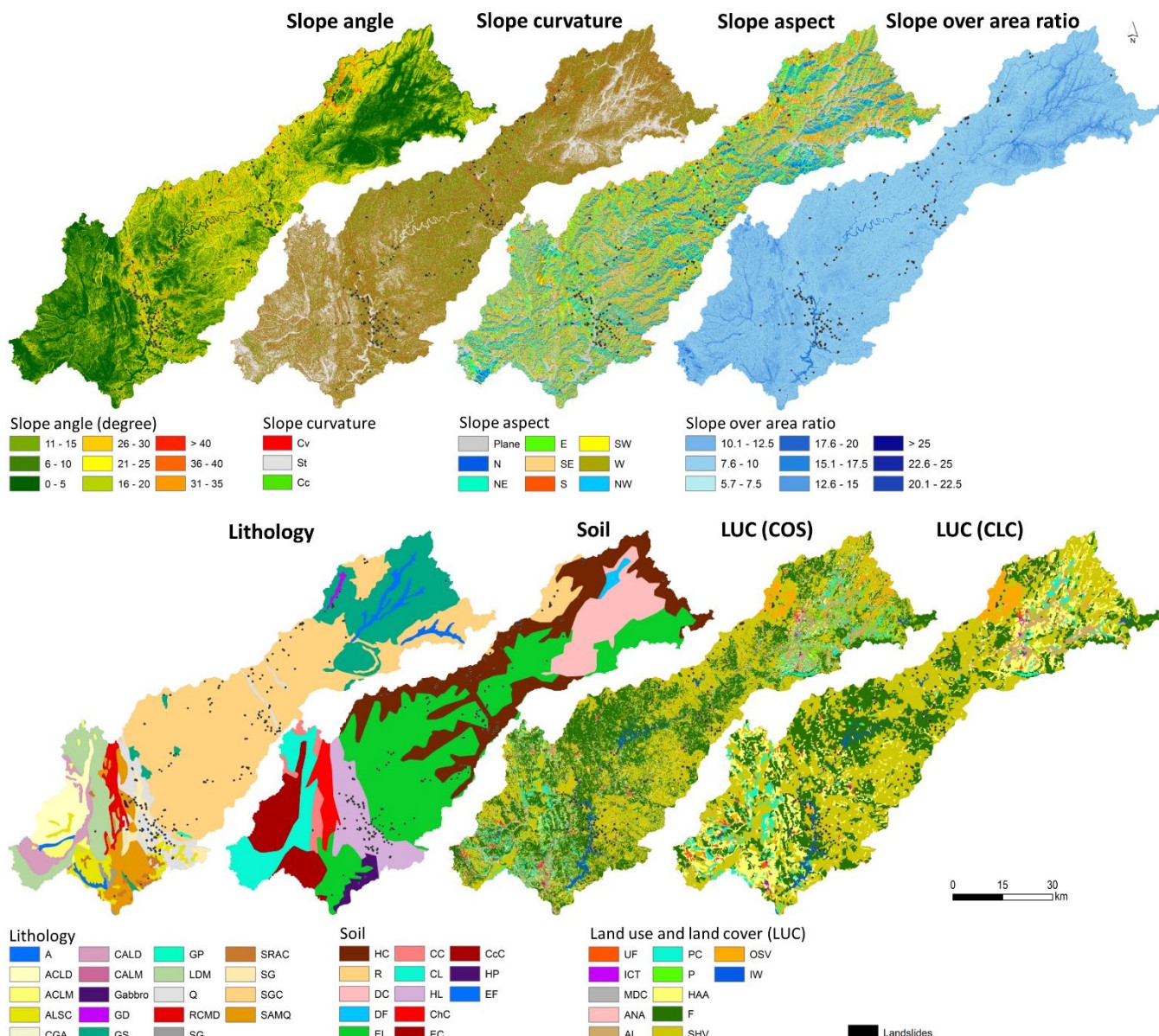

**Figure 2.** Predisposing factors used in the landslide susceptibility assessment.

**Predisposing factor maps legend**: **Curvature** - Cv: Convex, St: straight, Cc: concave; **Lithology** - A: Alluvium, ACLD: Arenites, conglomerates, limestones, dolomitic limestone,
ACLM: Arenites, conglomerates, limestones, dolomitic limestone and marl, ALSC: Arenites, limestone, sand, stony banks and clay, CGA: Clayey schist, grauwackes and arenites,
CALD: Conglomerates, arenites, limestone, dolomitic limestone, marly limestone and marl, CALM: Conglomerates, arenites, white limestone and red marl, G: Gabbro, GD: Glacial
deposits, GS: Granite and other stones, GP: Granite porphyritic, LDM: Limestones, dolomitic limestone, marly limestone and marl, Q: Quartzite, RCMD: Red sandstone,
conglomerates, marl and dolomitic limestones, SG: Sands and gravel, SRAC: Sands, rocky, arenites and clay, SG: Schists and grauwackes, SGC: Schists and grauwackes complex,
SAMQ: Schists, amphibolite, mica schists, quartzite grauwackes, carboned stones and gneisses; **Soil** - HC: Humic Cambisols, R: Rankers, DC: Dystric Cambisols, DF: Dystric
Fluvisols, EL: Eutric Lithosol, CC: Calcic Cambisols, CL: Calcic Luvisols, HL: Hortic Luvisols, ChC: Chromic Cambisols, EC: Eutric Cambisols, CcC: Calcic-chromic Cambisols,
HP: Hortic Podzols, EF: Eutric Fluvisols; **LUC** - UF: Urban fabric, ICT: Industrial, commercial and transport units, MDC: Mine, dump and construction sites, ANA: Artificial, non-
agricultural vegetated areas, AL: Arable land, PC: Permanent crops, P: Pastures, HAA: Heterogeneous agricultural areas, F: Forests, SHV: Scrub and/or herbaceous vegetation
associations, OSV: Open spaces with little or no vegetation, IW: Inland waters.

In general terms, an increasing slope angle promotes landslide occurrence and is a very good proxy of the shear stress (Zêzere et al., 2017). Slope instability is more frequent in the higher slope angles of the Estrela Mountain and throughout the Zêzere river valley. Also, in these areas, convex slope curvature is predominantly related to slope instability. The slope aspect is important in the spatial distribution of the different LUC types of the study area (Fig. 2) and on slope instability, especially in northwest-facing slopes (more exposed to rain and with higher humidity levels).

The SOAR is a proxy variable of the moisture retention, the soil water content, and the surface saturation zones (Zêzere et al., 2017), highlighting, in the Zêzere watershed, the upstream (very close to the Zêzere River) and south-west areas with a higher SOAR.

In the sample areas of the Vila de Rei and Ferreira do Zêzere municipalities, where a high landslide density was observed, schist and metasedimentary lithology are predominant. Further, slope instability in the watershed is higher in the hortic luvisols and in the LUC classes of forest and shrubland or herbaceous vegetation associations (Fig. 1).

The official LUC data available for the study area is the CLC produced by the European Environment Agency (EEA) and the COS produced by the General Directorate for Territorial Development (DGT) in Portugal. This LUC data (CLC and COS) has different properties and has been used in several studies about landslides in the Portuguese territory (e.g., Guillard and Zêzere, 2012; Meneses et al., 2015b; Piedade et al., 2011; Reis et al., 2003; Zêzere et al., 2017).

Table 1 describes the main properties of this LUC data (DGT, 2013; EEA, 2007; IGP, 2010). Among the differences between the two LUC datasets, the scale is highlighted because the COS is the most detailed relative to the CLC (proportion 1/4). However, the properties are not proportional between the two LUC datasets; while the COS features have a minimum mapping unit of 1 ha, the CLC has a minimum mapping unit of 25 ha; and the minimum distance between lines is 20 m in the COS, while in the CLC, it is 100 m. To reduce possible discrepancies in the field, the LUC data was collected for near dates: CLC 2006 and COS 2007. The LUC data was developed with base information that matches in temporal terms, for example, the satellite images, orthophotos, and agricultural and forestry inventories used as auxiliary information. The nomenclature of this LUC data corresponds to the third level (see the official CLC

nomenclature on the EEA website). In this study, the second level of the CLC nomenclature was used because it has a lower number of classes for the study area (12 of 31 classes, respectively).

**Table 1.** Properties of LUC data.

| Properties | Land Cover Maps of Portugal | Corine Land Cover |
|---|---|---|
| Acronym | COS | CLC |
| Scale | 1:25 000 | 1:100 000 |
| Minimum mapping unit | 1 ha | 25 ha |
| Data structure | Vector | Vector |
| Geometry | Polygons | Polygons |
| Minimum distance between lines | 20 m | 100 m |
| Base data | Orthophotos | Satellite images |
| Spatial resolution | 0.5 m | 20 m |
| Nomenclature | Hierarchical (5 levels) | Hierarchical (3 levels) |
| | 225 classes | 44 classes |
| Production method | Visual interpretation | Semi-automated production and visual interpretation |
| Date of production | 2007 | 2006 |

The agreement between the LUC data is presented in Table 2. The forest class shows great differences between the two LUC datasets. For example, the COS represents more forest area relative to the CLC (34 and 26.9 % of the study area, respectively), because a part of the COS (approximately 10 % of the study area) is classified as scrub and/or herbaceous vegetation associations in CLC. The reverse was also

verified; approximately 5 % of the study area is classified as scrub and/or herbaceous vegetation associations in the COS, and this same area is represented by forest class in the CLC. These discrepancies are derived from the LUC data properties because the COS is more detailed and represents more degraded forest areas, especially where wildfires occurred. These events affected a large percentage of the watershed (Meneses et al., 2018a), especially the Central sector, as a vast burned area culminated in a

large transition of forest area to shrubland.

The forest, scrub and/or herbaceous vegetation associations and open spaces with little or no vegetation are the LUC types predominant in the hillsides with steep slopes (see Tables 1 and 2 in the supplementary data). The remaining LUC classes present more area in the lower slopes (> 10 degrees).

The soil and lithology data were obtained from the Environment Atlas web platform, published by the

Portuguese Environment Agency (APA) at 1:1 000 000 scale. A digital elevation model (DEM) was built

using digital topographic maps at 1:25 000 scale (IGEOE), containing contour lines with 10 m equidistance.

**Table 2.** LUC data agreement (area ha) between CLC and COS classes.

| Data | COS | | | | | | | | | | | | |
|---|---|---|---|---|---|---|---|---|---|---|---|---|---|
| CLC | Urban fabric (UF) | Industrial, commercial and transport units (ICT) | Mine, dump and construction sites (MDC) | Artificial, non-agricultural vegetated areas (ANA) | Arable land (AL) | Permanent crops (PC) | Pastures (P) | Heterogeneous agricultural areas (HAA) | Forests (F) | Scrub and/or herbaceous vegetation associations (SHV) | Open spaces with little or no vegetation (OSV) | Inland waters (IW) | Total |
| UF | 3 160.2 | 439.8 | 77.3 | 100.8 | 207.7 | 502.0 | 15.7 | 929.2 | 337.7 | 251.5 | 0.1 | 18.7 | 6 040.7 |
| ICT | 134.1 | 650.4 | 83.0 | 9.5 | 33.4 | 27.4 | 9.0 | 62.5 | 130.8 | 207.7 | 0.3 | 8.1 | 1 356.1 |
| MDC | 6.1 | 58.3 | 283.0 | 0 | 3.6 | 3.6 | 6.8 | 6.5 | 48.2 | 53.5 | 0.2 | 5.4 | 475.0 |
| ANA | 29.3 | 2.9 | 0 | 22.5 | 0 | 0 | 0 | 0 | 1.7 | 9.1 | 0 | 0 | 65.6 |
| AL | 245.3 | 171.7 | 25.0 | 12.2 | 9 166.1 | 1 304.4 | 2 225.0 | 1 317.1 | 1 133.2 | 1 435.9 | 51.0 | 190.7 | 17 277.5 |
| PC | 1 271.4 | 93.3 | 37.3 | 21.2 | 1 357.9 | 7 948.5 | 315.4 | 2 930.0 | 2 004.5 | 2 300.2 | 7.9 | 38.1 | 18 325.7 |
| P | 4.4 | 2.4 | 0 | 0 | 61.3 | 0.9 | 36.1 | 58.4 | 41.2 | 188.6 | 0 | 0 | 393.2 |
| HAA | 7 791.6 | 736.5 | 271.4 | 73.7 | 11 773.1 | 15 553.2 | 2 341.0 | 23 762.4 | 16 514.4 | 12 935.5 | 143.3 | 243.9 | 92 140.0 |
| F | 745.3 | 392.9 | 173.1 | 29.3 | 741.9 | 1 715.5 | 238.1 | 4 058.7 | 100 486.5 | 26 805.7 | 42.0 | 735.8 | 136 164.8 |
| SHV | 826.5 | 510.0 | 259.3 | 38.0 | 1 353.1 | 2 543.2 | 958.3 | 5 832.8 | 50 509.8 | 149 644.0 | 4 052.8 | 846.7 | 217 374.5 |
| OSV | 29.4 | 13.8 | 5.3 | 1.4 | 18.3 | 10.3 | 10.7 | 140.4 | 860.0 | 6 367.1 | 4 206.6 | 30.3 | 11 693.7 |
| IW | 5.6 | 12.0 | 0 | 0.2 | 1.3 | 7.5 | 0 | 15.2 | 278.5 | 180.7 | 2.4 | 4 589.5 | 5 093.0 |
| Total | 14 249.1 | 3 084.1 | 1 214.7 | 308.8 | 24 717.7 | 29 616.3 | 6 156.0 | 39 113.2 | 172 346.6 | 200 379.5 | 8 506.6 | 6 707.1 | 506 399.7 |

Slope angle, slope aspect, slope curvature, and SOAR (topographic wetness index) layers were extracted from the DEM. Road network data (vector lines) were extracted from Portugal's military cartography (itinerary maps, 1:500 000 scale), available on the Portuguese Army Geospatial Information Center's website. The road network was classified according to the roads' width and their network hierarchy. Considering the road center line, a buffer of 5 m was defined for municipal roads, 10 m for complementary roads, and 20 m for motorways. These distances were measured with geographic information systems (GIS) on the study area roads (directly on the orthophotos).

The landslide inventory was obtained using photointerpretation (orthophotos from 2005 and Google Earth images), a process supported by the ancillary topographic data and further fieldwork validation only performed in the sample areas (Fig. 1) due to the extension of the study area. A total of 128 landslides (predominantly shallow translational slides), with a total area of 74 042 m$^2$, was validated during fieldwork in the sample areas (49.4 % of the total inventoried landslide cases). Among the landslides initially inventoried by photointerpretation in the sample areas, more than 90 % of cases were confirmed. In these sample areas, road disruptions were also validated.

For the complete Zêzere watershed, 259 landslides were identified, predominantly of shallow type. Of the total, 32 landslides directly affected the road network (total or partial blockages by the material and 7 cases with partial loss of infrastructure). The landslide inventory was randomly divided into two subsets (Fig. 1) (Chung and Fabbri, 2003): the landslide training group and the landslide test group (81.5 % and 18.5 % of the total landslide affected area, respectively). The statistical description of each landslide group is presented in Table 3.

**Table 3.** Statistics description of the training group and test group landslide inventories.

| | Training group | | Test group | | Total inventory |
|---|---|---|---|---|---|
| | Non affected roads | Affected roads | Non affected roads | Affected roads | |
| Total landslides | 185 | 26 | 42 | 6 | 259 |
| Total area (m$^2$) | 44 604 | 369 404 | 10 444 | 12 089 | 104 077 |
| Minimum (m$^2$) | 134 | 7 | 18 | 82 | 7 |
| Maximum (m$^2$) | 27 364 | 12 507 | 1 911 | 5 881 | 12 507 |
| Mean (m$^2$) | 2 414 | 1 421 | 249 | 2 015 | 402 |
| Standard deviation (m$^2$) | 3 284 | 2 647 | 304 | 2 627 | 1 069 |

The landslide size frequency distribution is different between the landslides that affected the road network and those that did not (Fig. 3). The area of the majority of landslides ranges between 101 and 200 m$^2$, while most of the landslides that affected the road network present a larger area (> 1 000 m$^2$).

All the predisposing factors and landslide inventory were converted to raster (resolution 10 m) to assess the landslide susceptibility. The selection of the predisposing factors' cell size was based on several geoinformation conversion tests in the Zêzere watershed previously performed by Meneses et al. (2016, 2018b).

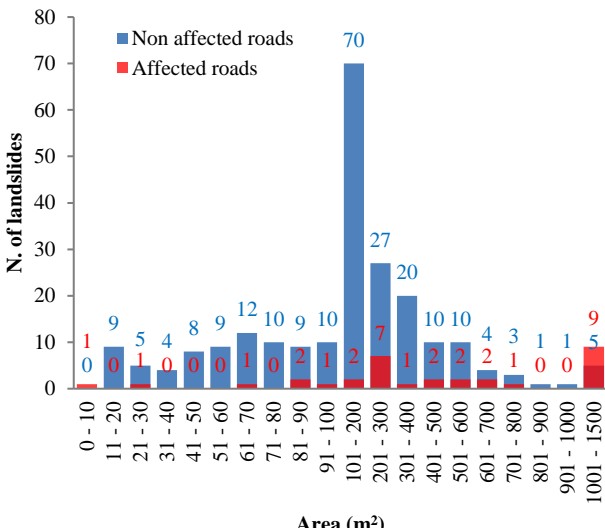

**Figure 3.** Landslide size frequency distribution.

## 2.3 Methods

The landslide susceptibility modeling was carried out using the Information Value (IV) method (Yan, 1988; Yin and Yan, 1988). The IV method is a bivariate statistical method that has been used in several studies and different areas with good results for landslide susceptibility assessment (e.g., Guillard and Zêzere, 2012; Oliveira et al., 2015a; Zêzere et al., 2017). The IV of each class within each explanatory variable is given by Eq. (1) (Yan, 1988; Yin and Yan, 1988):

$$IVx_i = \ln \frac{S_i / N_i}{S / N} \tag{1}$$

where $IVx_i$ is the IV of the variable $x_i$; $S_i$ is the number of terrain units with landslides and the presence of variable $x_i$; $N_i$ is the number of terrain units with variable $x_i$; $S$ is the total number of terrain units with landslides, and $N$ is the total number of terrain units.

The IV method was applied in several landslide susceptibility zonation studies, providing good results (e.g., Che et al., 2012; Chen et al., 2016; Conforti et al., 2012) at the regional scale. This method was also applied in several studies conducted in the Portuguese territory, with good performance in susceptibility

assessment (e.g., Guillard and Zêzere, 2012; Oliveira et al., 2015b; Pereira et al., 2014; Zêzere et al., 2017).

The *a priori* probability of finding a landslide unit in the study area ($S/N$) and conditional probabilities for each class of the independent variables ($S_i/N_i$) were calculated, obtaining the IV for these classes. However, the IV method presents constraints on obtaining the natural logarithm for negative results; in this case, the lower value calculated for each variable was assigned to classes when $S_i$ is equal to zero. The IV of all the variables were combined to obtain the landslide susceptibility map (LSM). For the final landslide susceptibility assessment, i.e., the integration of the IVs of all the independent variables, the following equation was considered:

$$IV_j = \sum_{i=0}^{n} X_{ij} I_i \qquad (2)$$

where $IV_j$ is the total IV of the cell $j$, $I_i$ is the information value of each cell of each independent variable, $n$ is the number of variables, $X_{ij}$ assumes the value 1 or 0, depending on the presence or absence of the variable in the terrain unit.

Landslide susceptibility model performance was assessed using training landslides. Landslide areas in the test group were only used to perform an independent validation of the landslide susceptibility. Prediction rate curves (PRC) were computed for each final LSM (Chung and Fabbri, 1999, 2003) and also the area under the curve (AUC). Success rate curves (SRC) were obtained for the landslide susceptibility road network maps using only the landslides that affected roads.

The importance of each independent variable in the landslide susceptibility assessment was also determined, so that the spatial influence of each predisposition factor in the models can be understood. The accountability ($A_I$) and reliability ($R_I$) indexes have been used in different contexts to assess the importance of each independent variable in the bivariate statistical methods (e.g., Blahut et al., 2010; Meneses et al., 2016). $A_I$ explains how different classes of predisposition factors are relevant in the analysis because they contain the landslide area, while $R_I$ depends on the average density of the landslide area in the predisposing factors classes that are more relevant to the development of this process. In this procedure, the $A_I$ and $R_I$ were determined using Eq. 3 and 4, respectively (Blahut et al., 2010).

$$A_I = \frac{\sum_{i=1}^{n} k}{N} 100 \tag{3}$$

$$R_I = \frac{\sum_{i=1}^{n} k}{\sum_{i=1}^{n} y} 100 \tag{4}$$

where $k$ is the landslides area in classes with the conditional probability values higher than a priori probability; $N$ is the total landslides area; $y$ the area of each class of independent variable with a conditional probability above the a priori probability.

Two landslide susceptibility models were built using the IV method (see results in Table 3 in the supplementary data), using the same set of predisposing factors, except the LUC data (Fig. 4): model 1 (M1) was modeled with the COS 2007 and resulted the landslide susceptibility map 1 (LSM1); model 2 (M2) was modeled with the CLC 2006 and resulted the landslide susceptibility map 2 (LSM2). LSM1 and LSM2 were correlated, and the corresponding spatial agreement was analyzed.

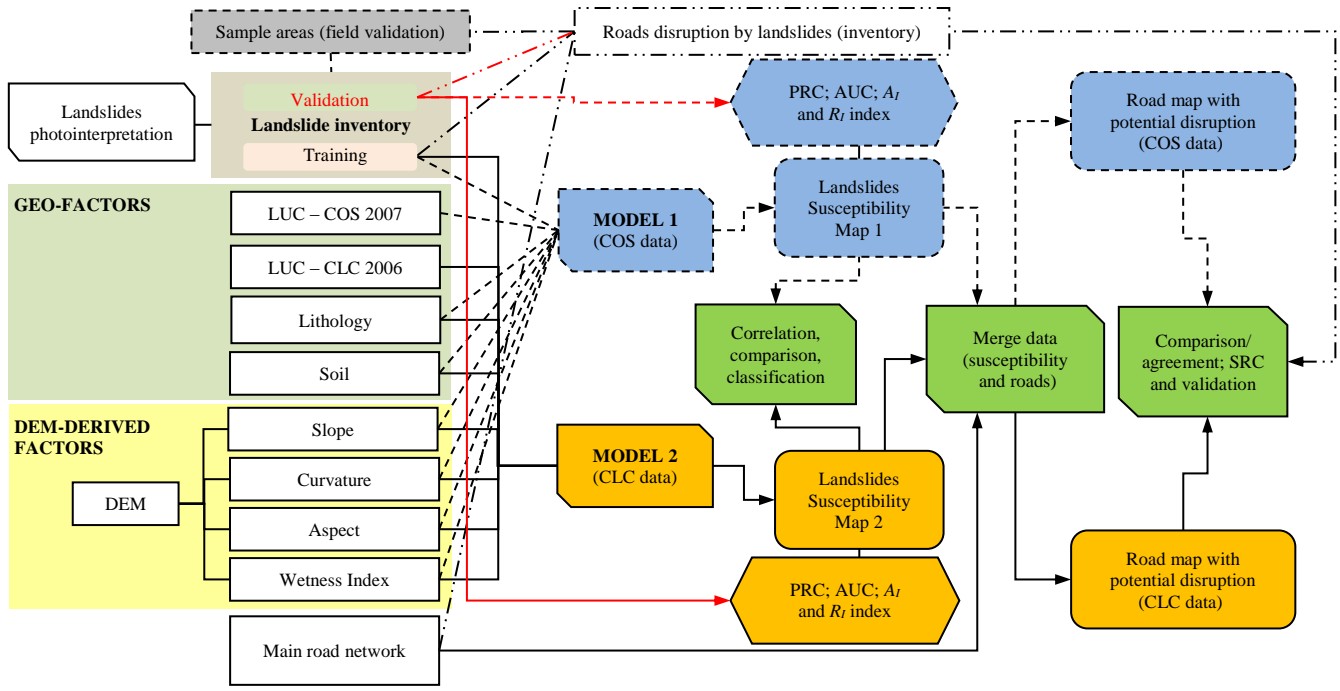

**Figure 4.** Workflow of landslide susceptibility assessment (using different LUC datasets) and the roads susceptibility data integration.

Information values of LSM1 and LSM2 were assigned to the road network (using GIS), resulting in a road network map with the landslide susceptibility location (landslide susceptibility of the roads network - LSRN1 and LSRN2, respectively), where there is a higher spatial probability of road interruption or road interference caused by landslides. Different outputs of the two models (road network) were compared using the overall agreement and Kappa coefficient (Congalton and Green, 2009), allowing the assessment of the consistency and agreement of the obtained results with different LUC datasets. The information of road disruptions caused by landslides were used to validate these results.

Landslide susceptibility maps were built and classified in 10 classes (deciles) containing an equal number of terrain units to allow visual comparison of the results.

# 3 Results

## 3.1 Landslide susceptibility

The landslide susceptibility results show spatial contrasts in the study area. Some areas in the center of the watershed (highlighting the vicinity of the Castelo de Bode reservoir) and the northern sectors (highlight the Estrela Mountain) present the highest landslide density and susceptibility (Fig. 5).

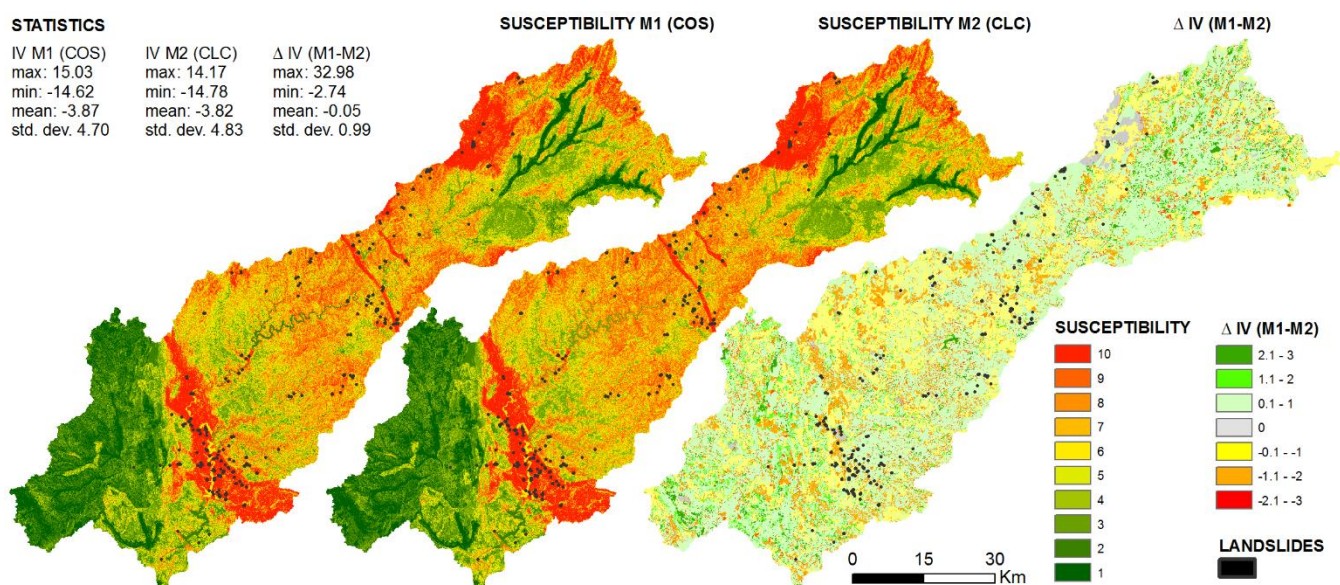

**Figure 5.** Landslide susceptibility (IV represented from the highest (red) to the lowest susceptibility (green)): the map of Susceptibility M1 represent the results obtained with model 1 (performed with COS data) – LSM1; the map Susceptibility M2 represent the results obtained with model 2 (performed with CLC data) – LSM2. The map in the right is the variation between LSM1 and LSM2.

The results of the $A_I$ and $R_I$ indexes show important differences between the predisposing factors that have integrated the landslide susceptibility models (Table 4). The LUC predisposing factors (the COS and CLC) registered the highest $A_I$ results, highlighting the COS's LUC with a higher $A_I$. These results show the relevance of certain classes of the COS in the predisposing factors dataset, by the number of landslide areas covered (emphasis on the forests, scrubland, and/or herbaceous vegetation associations and open spaces with a scarcity or absence of vegetation).

**Table 4.** Results of the accountability ($A_I$) and reliability ($R_I$) indexes.

| Factors | $A_I$ | $R_I$ |
|---------|-------|-------|
| Aspect | 79.5 | 0.2 |
| Slope | 76.1 | 0.6 |
| SOAR | 13.5 | 0.7 |
| Soil | 62.4 | 1.0 |
| Lithology | 60.6 | 0.4 |
| Curvature | 61.1 | 0.3 |
| LUC (COS) | 82.0 | 0.3 |
| LUC (CLC) | 76.0 | 0.3 |

The soil, SOAR, and slope angle present the highest values in the case of $R_I$, which shows that landslide density is concentrated in a reduced number of classes of each of the predisposing factors areas (e.g.,

Hortic Luvisols, SOAR [22.5-25], and slope [between 25° and 45°]).

The landslide susceptibility model's agreement test was performed using the landslide training inventory used to perform the outputs of each landslide susceptibility model, and these results were validated using the landslides test group. The PRC of each final susceptibility map (obtained from the results of the landslide training group) show slight variations (Fig. 6), but, in general terms, the curves are identical,

demonstrating the high and similar performance of the models in the determination of landslides susceptible areas.

The AUC of LSM1 and LSM2 that includes the same landslide information used to train the models is 94.1 % and 93.9 %, respectively. These results (landslide prediction) were considered to integrate the landslide susceptibility road network (LSRN1 and LSRN2) and the next analyses presented. Additionally,

spatial differences were observed in the landslide susceptibility maps (Fig. 5), reflecting the differences of the influence of LUC properties.

When the two landslide susceptibility maps are reclassified into two classes (not susceptible IV ≤ 0 and susceptible IV > 0), the susceptible area in LSM1 corresponds to 19.7 % and in LSM2 to 20.8 %. The CLC data provide IV results lower than the IV obtained with the COS data, but the CLC is more

generalized and justifies that the most susceptible area observed in LSM2, compared to LSM1. The variation between the maximum and minimum IVs (3 and -3 of ΔIV in Fig. 5) show the landslide susceptibility differences derived of spatial representation of LUC classes of the two LUC datasets considered. The highest variations between LSM1 and LSM2 are found in places with reduced IVs (low

and moderate susceptibility), marking the central sector of the study area. The areas with the highest IVs in LSM1 and LSM2 present a lower variation.

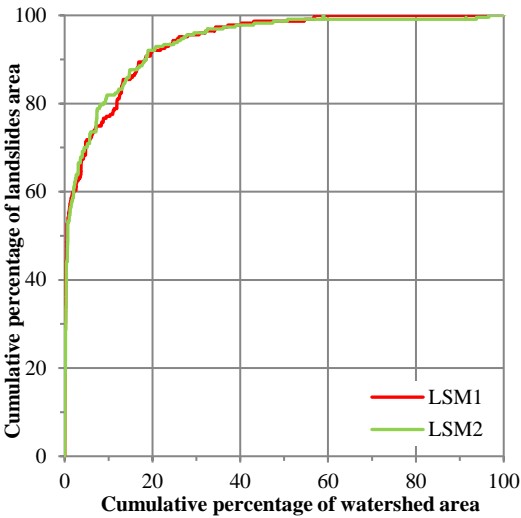

5    **Figure 6.** Prediction rate curves (PRC) of the landslide susceptibility (LSM1 – COS and LSM – CLC).

## 3.2 Landslides susceptibility in the road network

Due to the width of the road network, in most cases, these infrastructures are not identified in the LUC data due to the properties or specifications (Table 1), namely, the minimum distance between lines considered in each LUC data in the research. The class "road and rail networks and associated land" (LUC nomenclature, level III) integrates the main class "industrial, commercial, and transport units" (level II); however, when a tabulation of the area of the roads network used in this research and the LUC datasets was performed, the density of the roads was different in each LUC class between datasets (Fig. 7).

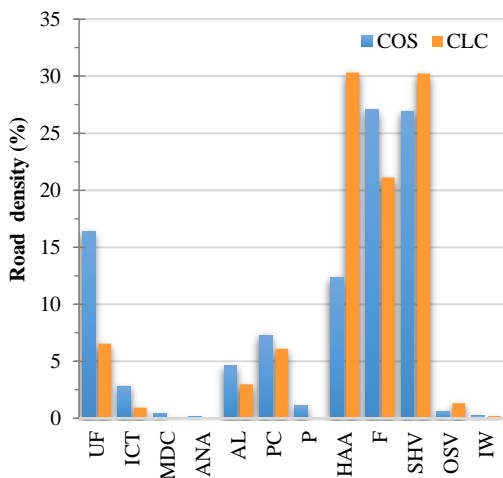

**Figure 7.** Density of roads by LUC class of CLC and COS data (see LUC legend in Fig. 2).

The IVs of LSM1 and LSM2 assigned to the road network differentiated the roads according to the landslide susceptibility, representing the highest IV where future landslides will occur and possibly rupture of the road network or cause socioeconomic constrains due to total or partial blockages. In this case, the differences of the roads landslide susceptibility were also analyzed.

The IVs assigned to the road network do not have spatial agreement between the two models. The difference between the maximum and minimum IVs of the LSRN1 and LSRN2 variations is notorious, with approximately 1 value of IV variation. The interquartile range of the IV is greater in LSRN2 than in LSRN1 (Fig. 8). However, the IV average is similar in LSRN2 in comparison to LSRN1.

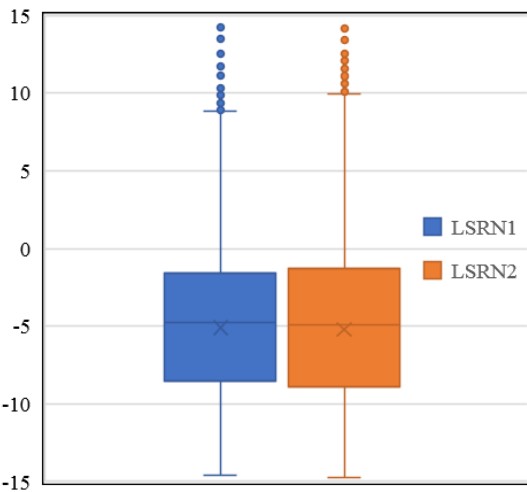

**Figure 8.** Landslide susceptibility of the road network. LSRN1 – IV assigned for the LSM1; LSRN2 – IV assigned for the LSM2.

Landslide susceptibility map of the road network obtained by LSM1 (resulting in LSRN1) (Fig. 9) shows
that it is spatially contrasted along the road network, highlighting the places where future landslides that
may cause disturbances on the roads are most likely to occur. On the other hand, in the landslide
susceptibility map of the roads network obtained by LSM2 (resulting in LSRN2), the IV assigned to the
road network is generally lower when compared to LSRN1, a result derived from the LUC generalization
(CLC) used in the input of model 2.

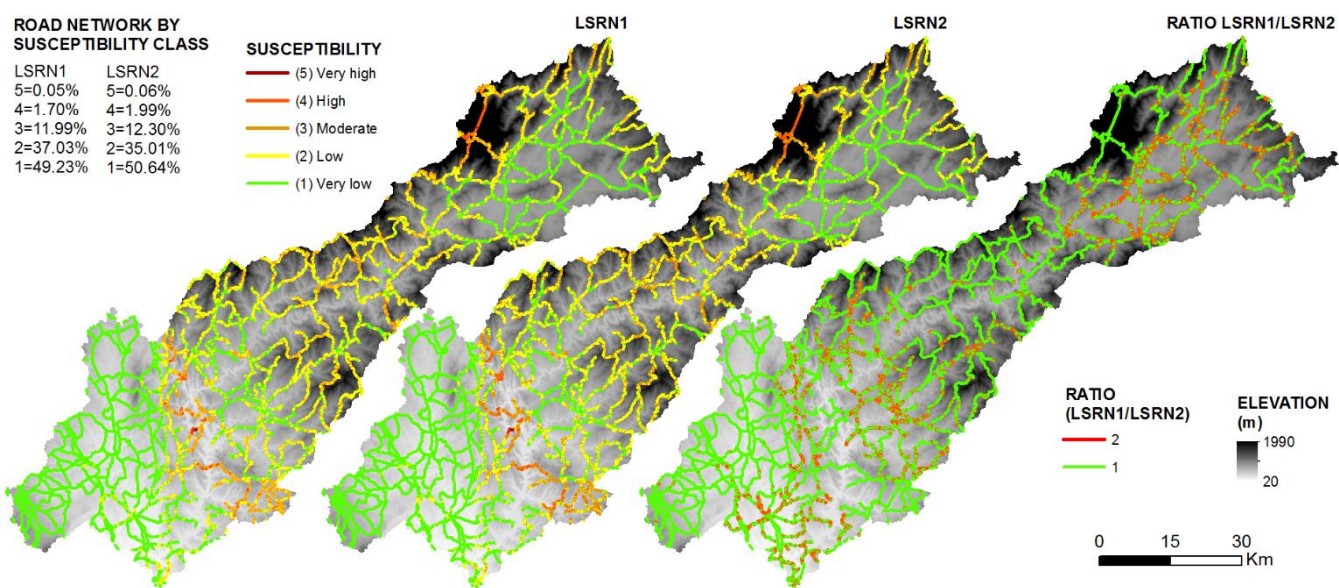

**Figure 9.** Landslide susceptibility of the road network (LSRN1 and LSRN2) and the ratio between landslide susceptibility class of the roads. LSRN1 – IV assigned of the LSM1; LSRN2 – IV assigned of the LSM2.

LSRN1 includes 14.1 % of the roads with a positive landslide susceptibility (IV ≥ 0), and the roads with high landslides susceptibility (IV > 10) represent only 0.1 % of the total road network (Fig. 9). In LSRN2, the positive landslide susceptibility (IV ≥ 0) increases (compared with LSRN1) and comprises 14.7 % of the total road network, where 0.1 % of this network corresponds to a high landslide susceptibility (IV > 10).

LSRN2 does not show a high variation in short roads distances, i.e., the IV tends to be extended within each polygon of the same class of the CLC's LUC (larger polygons in comparison with the COS data), reducing the IV variation along the roads. The variation of the IV within each polygon of the LUC data is only explained by the remaining predisposing factors included in the model.

In LSRN2, the places with a high landslide susceptibility are not always identified as those where

landslides effectively occurred (Fig. 10). The landslide susceptibility of the road network enhances the results obtained with the COS (LSRN1) in very high landslide susceptibility areas, precisely where landslides were validated in the fieldwork. These results show the importance of LUC data properties in the spatial differentiation of landslide susceptibility.

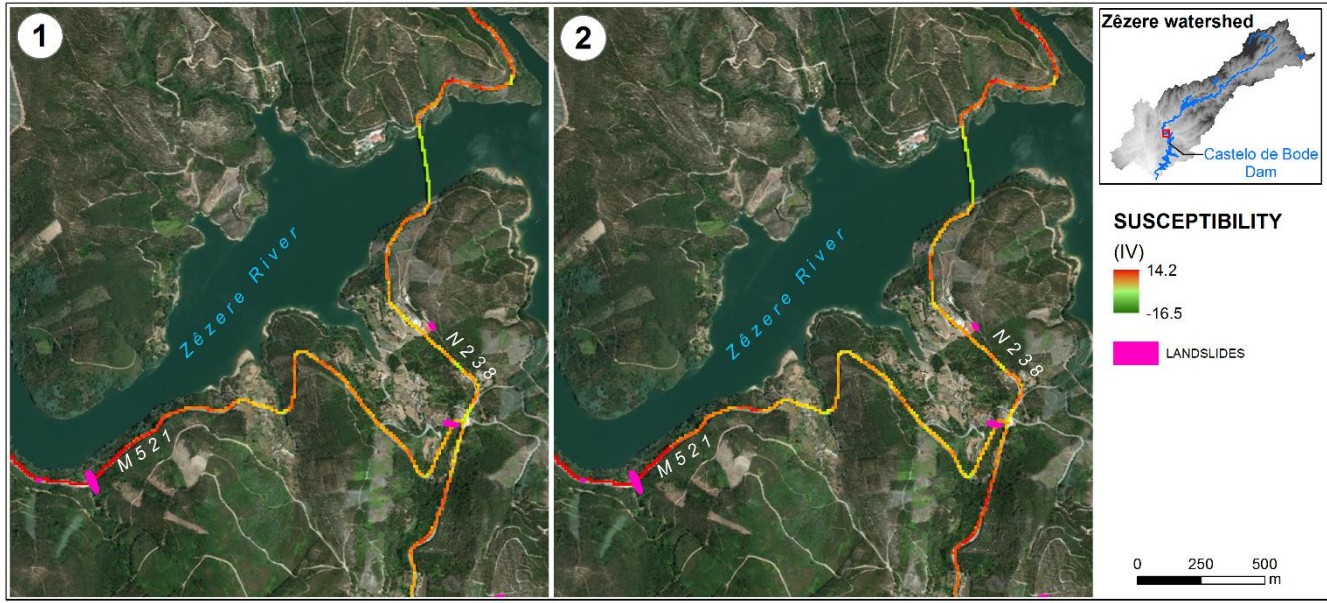

**Figure 10.** Examples of the landslide susceptibility of the road network in Ferreira do Zêzere municipality. 1 – LSRN1; 2 – LSRN2.

The spatial agreement and Kappa coefficient between the LSRN1 and LSRN2 landslide susceptibility
classes are 89.7 and 83.1 %, respectively (Table 5). In general, the individual susceptibility classes present
a high agreement ($\geq$ 80 %, except the high and very high classes of LSRN2) but with differences between
the two models. For example, the landslide susceptibility class "very high" comprises 0.05 % and 0.06 %
of the total road network in LSM1 e LSM2 respectively, but LSRN2 presents 20.4 % of the omission
differences in same susceptibility class compared to the 3.8% commission differences of LSRN1. The
intermediary susceptibility classes of the two models highlight the omission and commission differences.
Although variations exist between LSRN1 and LSRN2 landslide susceptibility, the relationship between
the two models' outputs is high, presenting a Pearson correlation coefficient of 0.98 (significance level
$p < 0.05$). The results of this correlation reflect the existence of an agreement on the spatial variation
between LSRN1 and LSRN2, i.e., in general, when the IV of one output increases the other also increases,
or vice versa, regardless of the discrepancy between the IVs of the same cells of each output.

**Table 5**. Spatial agreement between LSRN1 and LSRN2 (% of road network).

| LSRN1 | LSRN2 Very low (IV < -5) | Low (IV -5-0) | Moderate (IV 0-5) | High (IV 5-10) | Very high (IV >10) | Total area (%) | Agreement (%) | Commission differ. (%) |
|---|---|---|---|---|---|---|---|---|
| Very low (IV < -5) | 46.3 | 2.2 | 0.0 | 0.0 | 0.0 | 48.5 | 95.5 | 4.5 |
| Low (IV -5-0) | 3.5 | 31.7 | 2.3 | 0.0 | 0.0 | 37.4 | 84.6 | 15.4 |
| Moderate (IV 0-5) | 0.0 | 1.7 | 10.1 | 0.5 | 0.0 | 12.3 | 82.4 | 17.6 |
| High (IV 5-10) | 0.0 | 0.0 | 0.2 | 1.5 | 0.01 | 1.7 | 88.1 | 11.9 |
| Very high (IV >10) | 0.0 | 0.0 | 0.0 | 0.0 | 0.05 | 0.05 | 96.2 | 3.8 |
| Total area (%) | 49.8 | 35.5 | 12.58 | 2.03 | 0.06 | | | |
| Agreement (%) | 93.0 | 89.2 | 80.3 | 75.0 | 79.6 | Overall agreement: 89.7% | | |
| Omission differ. (%) | 7.0 | 10.8 | 19.7 | 25.0 | 20.4 | Kappa coefficient: 83.1% | | |

The LSRN1 and LSRN2 results were crossed with all landslides that caused perturbations or disruptions of the road network, and the performance of models was assessed. Overall, the results were very good, with 89.5 and 89.3 % AUC for LSRN1 and LSRN2, respectively. However, LSRN1 offers slightly better results when compared to LSRN2, as it can be seen in the representation of the SRC (Fig. 11), i.e., up to 20 % of the total area of the road network validates approximately 83 % of the landslide susceptibility of LSRN1 and LSRN2. Nevertheless, the LSRN2 shows a slightly better performance (to approximately 45 % of the total area of the road network), but the LSRN1 improves its validation performance at this point, being completely validated with 67 % of the total area of the road network, while LSRN2 is validated with 74 % of its area.

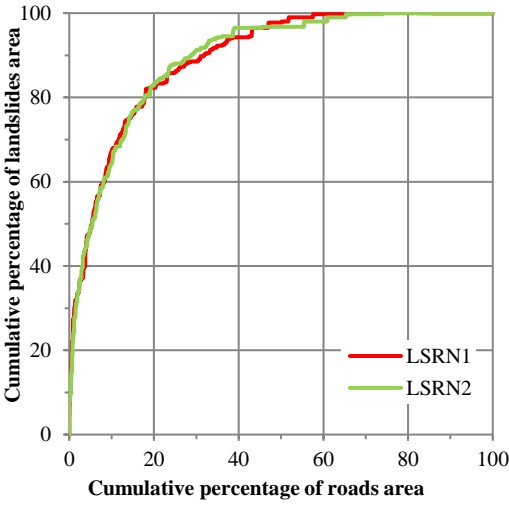

**Figure 11.** Success rate curves of LSRN1 and LSRN2 models.

**4 Discussion**

In landslide hazard and risk assessment, the LUC data integrate the controlling factors group and, in many evaluations, is pointed by another factor input to the model. Usually LUC data is used as a landslide conditioning factor which, in some cases, is scarce, generalized and low detailed. For example, Eeckhaut and Hervás (2012) verified that in the different locations of Europe the CLC is widely used for landslide assessment, because is the only LUC data available. Remote sensing and satellite images contributed to LUC data acquisition for landslide susceptibility assessment in different times (Guzzetti et al., 2012) and territories, and minimize some problems of scarcity and detail (thematic and resolution). LUC is an important conditioning factor in landslide susceptibility (Pisano et al., 2017), and the high accountability index results prove this fact (Table 4).

There are several studies about the influence of land use cover changes on landslide susceptibility (e.g., Karsli et al., 2009; Mugagga et al., 2012; Promper et al., 2014; Reichenbach et al., 2014), although to the best of our knowledge there are no approaches that analyze the influence of different LUC datasets with different properties (date and base maps used on the production, spatial resolution, scale, minimum mapping unit, or others) on the landslide susceptibility results. When the landslide predisposing factors are collected, the LUC dataset must be selected according to its abovementioned properties, and not only on the basis of its availability and free of charge conditions.

When different LUC datasets are available, the choice for the LUC dataset used in the landslide susceptibility assessment is not always clearly justified, and the results may vary according to LUC data properties selected. For Portuguese territory different LUC datasets (with different properties) are available, but the use of each dataset can generate different conclusions, for example, different land use and land cover changes in the same period were observed by Meneses et al. (2018c).

This study highlights the landslide susceptibility differences derived exclusively from the LUC data properties, because the other predisposing factor maps are the same in both models. Although, if another method is used, the terrain mapping unit or other characteristic is changed, the results may vary, which has already been widely discussed (Chen et al., 2016; Den Eeckhaut et al., 2010; Guzzetti et al., 2006; Oliveira et al., 2015a; Zêzere et al., 2017).

Further, the data of soil and lithology was constrained and very generalized (1:1 000 000 and 1:50 000 scales, respectively), and this factor can influence the IV results if more detailed data was considered in the modeling process. The performance of the landslide susceptibility mapping and assessment is controlled by the quality of the available data, not only on the method (Pourghasemi et al., 2014).

5 Some research works refer to the quality of geoinformation (scale and precision) on the final results changes (e.g., Etter et al., 2006). In this case, the degree of completeness, and the positional, geometric, and thematic agreement of the selected LUC data was evaluated by different proprietary institutions, with more than 80 % accuracy, i.e., where the semantic inconsistencies error was reduced, an important factor in reducing the error propagation and achieving a product with best quality (Van Oort and Bregt, 2005; 10 Regnauld, 2015).

The landslide inventory was obtained by photointerpretation, which is certainly not complete, especially in forest and agricultural areas, a fact that could have impact on the landslide susceptibility zonation of the study area. This inventorying method does not allow for shallow or small landslide identification in forest areas, where the type, height, and density of the vegetation is important to landslide activity 15 (Guzzetti et al., 2012), or in cultivated areas where agricultural practices erase the morphological and LUC signature of slope failures (Fiorucci et al., 2011). The quality and completeness of the landslide inventories can interfere with the quality of future landslide spatial occurrences (Galli et al., 2008; Guzzetti et al., 2012; Reichenbach et al., 2018). However, the landslide inventory is the same for both landslide models presented in this research, and the variation results depend exclusively on the LUC 20 datasets that integrated each model.

The correlation between the outputs of each model is high, but there are spatial differences between them. The COS data is more detailed (1:25 000), than the CLC data (1:100 000), and the LUC classes are more differentiated in the territory, allowing greater detail and agreement in determining the areas with high landslide susceptibility, which were verified in LMS1; while in model 2, CLC data is less detailed and 25 contributes to IVs more reduced (low and very low landslide susceptibility in LSM2) compared with LSM1.

IV is more generalized along the road network at LSRN2 when compared with LSRN1, results derived exclusively from the input of LUC data with different properties in the models. These results highlight the importance of generalization/scale of LUC data selected in the landslide susceptibility assessment.

In the road network intersection with the LUC data, a high absence of roads data was observed in the class "industrial, commercial, and transport units," which is explained by the cartographic generalization due to the minimum mapping unit and minimum distance between lines of each LUC dataset. These factors exclude the roads data due to the minimum requirements defined in the technical specifications of each LUC dataset creation. However, the distribution of the road network between the LUC classes is quite variable in both LUC datasets (the COS and CLC), one of the factors that also justifies the variation of landslide susceptibility observed in different outputs.

The results of the PRC and AUC for LSM1 and LSM2 show a high quality and performance of both models in the landslides susceptibility areas determination (Guzzetti et al., 2006), but LSM1 presents a slightly better performance. Nevertheless, the prediction landslide results were validated with the landslide test group and present good results to be assigned in the road network.

The LSRN1 and LSRN2 models' validation results demonstrate that the models effectively identify the places where the landslides occurred and are more likely to occur in the future. In this case, the SRC and AUC note the high efficiency of the models (Guzzetti et al., 2006) with LSRN1 having a slightly higher efficiency, highlighting the properties of the LUC data.

Some roads in the study area were affected by landslides, a fact confirmed during the fieldwork developed to validate the landslide inventory (examples of some roads blockage or damaged in the Estrela Mountain and sample areas). In certain cases, the affected roads are important accesses to the most isolated villages in the study area and, in some cases, a landslide can isolate the villages because part of the affected infrastructures are unique public accesses, a fact verified in the sample areas.

The results highlight the importance of LUC data properties in landside assessment. More detailed LUC data (COS data) allows better landslide susceptibility results, a fact that was also described by Dymond et al. (2006), identifying some places where landslides occurred in the study area. Detailed predisposing factors data is recommended in landslide susceptibility assessment, a fact also mentioned in other studies,

e.g., Fressard et al. (2014) refer to the importance of detail in geomorphological variables to obtain high-quality results in landslides prediction.

This study was performed in a specific watershed, which highlights that landslide susceptibility changes according to the LUC data properties. It is recommended that the LUC data to be used as a predisposing factor of landslide susceptibility (e.g. in road networks) to be more detailed as possible, and avoiding small scale LUC datasets ($\geq$ 25 000). Further research is needed to test if these results change when the scale is different (e.g. national scale, or very detailed scale).

In the analysis of the risk associated with road transportation, the higher probability of a given event or incident, the greater the consequences (Berdica, 2002). In this context, the determination of the locales with the highest landslide susceptibility is very important, enabling prevention and minimizing these consequences, or to enabling better reactions when dealing with emergencies, because road closures change the traveling and reaction time (Meneses and Zêzere, 2012).

## 5 Conclusion

Landslide susceptibility in the Zêzere watershed is spatially variable, highlighting some characteristics of the study area's geo-factors in the high landslide density in a specific location, for example, the highest slope angles and certain LUC types (e.g., forests and scrubland) and lithology.

The properties of the data that integrates the landslide susceptibility models are also an important issue to be considered, since the variation of the properties of the same geo-factor provided different results, in this case, LUC with different properties.

More detailed LUC data (COS) allows better landslide susceptibility results, while more generalized LUC data (CLC) resulted in the landslide susceptibility more reduced, disallowing the identification of some places where landslides occurred. However, the results of the two susceptibility models showed a good performance, a fact demonstrated by the validation of the models' results.

The assignment of the landslide susceptibility results to the road network allowed the identification of the locations with the highest spatial probability for landslide occurrence. The LSRN1 map stands out with better results due to the integration of the COS dataset, showing the importance of LUC data detail in the identification of locations where landslides have occurred. The LSRN2 map does not have a good

performance in the identification of high landslide susceptibility in all road sections where landslides have occurred. In general, both LSRN1 and LSRN2 show the same trend in the spatial variation of landslide susceptibility of the study area's road network, highlighting the high susceptibility on the slopes of the Estrela Mountain and near the Castelo de Bode reservoir.

Finally, LUC data properties were shown to be important in the variation of landslide susceptibility results. Knowing the locations where landslides are likely to occur, alternatives options can be created to avoid partial or complete isolation of certain localities, reduce the social and economic constraints of this population, and adopt preventive measures and alternative evacuation paths in case of landslide occurrence.

## Acknowledgments

This work was financed by national funds through FCT-Portuguese Foundation for Science and Technology, I.P., under the framework of the project BeSafeSlide - Landslide early warning soft technology prototype to improve community resilience and adaptation to environmental change

(PTDC/GES-AMB/30052/2017), and by Research Unit UID/GEO/00295/2013 (Centre for Geographical Studies). Bruno Meneses was financed through a grant of the Institute of Geography and Spatial Planning and Universidade de Lisboa, IGOT-UL (BD2015).

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
