# Peer review of "Effects of different land use and land cover data on the landslide susceptibility zonation of road networks"

_Natural Hazards and Earth System Sciences, 2017_

## Referee Comment (RC1) · Anonymous Referee #1 · 14 Mar 2018

General comments

The manuscript addresses the effect of using land cover data of different spatial and thematic resolution in landslide susceptibility modeling, particularly for susceptibility zonation of a road network. The topic is interesting and significant. Land cover data is used in susceptibility mapping, often without questioning its quality and suitability for such an analysis. The authors provide us with an overview of what we might be missing in case we use unsuitable data (or data which is too coarse for example).

The authors first introduce us to landslides in general, and their effect on human lives, activities, infrastructure, etc. They proceed with describing the usefulness of landslide

susceptibility assessments. Afterwards, we are introduced to the role of land cover and how the choice of geoinformation details of land cover is usually not studied – despite being a significant factor. The authors demonstrate their approach in a watershed in Portugal (with a detailed focus on three smaller areas within the watershed boundaries). They use two different land use and land cover data to demonstrate the effect of using different data on landslide susceptibility: the Portuguese land cover map (COS), and the European Corine Land Cover (CLC). The landslide susceptibility mapping itself is straightforward, and is based on acknowledged and commonly used methods (information value). Although, there are other approaches, where similar data could be used (logistic regression, weights of evidence...), the authors chose this method, as it has been applied at a similar scale, also in Portugal. Generally, it is a nice study, however with some major flaws - most of them related to the fact that some topics were not addressed. This means, that my revision recommendation is mostly based on rewriting the main body of the text, adding additional clarifications, or expanding the discussion. Aditional analyses are not needed.

First, while the authors investigated the role of land cover data on modeling landslide susceptibility, they did not compare different methods. I do not expect the authors to perform additional analyses with other modeling approaches – however, I would like them to discuss the method used a bit more extensively. For example, could other methods lead to larger (or smaller) differences between different land cover data?

Second, I have seen many landslide susceptibility studies, where data with relatively coarse data has been used. Also here, the data on soil and lithology is on a (much) coarser resolution than other data (slope characteristics, and land cover). We can see, that while all other data has a fine and detailed pattern, the lithology and soil maps have clear boundaries, with relatively large mapping units. This is of course not the authors fault – this are probably still the most detailed soil and lithology maps available for the study area. Nevertheless, I would like to see a more detailed section in the discussion, reflecting on the discrepancy of such differences (e.g. scale and mapping

unit) and the effects on susceptibility modeling. The authors already did this for the land cover maps, and wrote a few sentences in the discussion.

Third, I have two comments regarding the landslide inventory (page 9, lines 5-10). To be clear, the authors mapped landslides using orthophotos and google earth themselves? And the authors went on the field themselves as well? Currently, it is not clear if they received the data that was developed by photointerpretation, or if they performed it themselves. Moreover, it seems that the most landslides are outside forested areas – I compared the two land cover maps with the landslide distribution map visually. This is of course possible, as evidence shows that forests have a positive role on slope stability (e.g. due to roots). Nevertheless, it is also difficult to map landslides in forests, using photointerpretation only. This can have significant effects on the results. For example, if we look at Figure 4, the areas where the differences between the two susceptibility maps are the lowest are indeed areas covered by forests (or seem to be, the authors did not provide additional information that would lead to other conclusions). Also, studies have shown that landslide unit definition have a significant effect on landslide susceptibility modeling. It makes a difference if a landslide is mapped as a point, as the whole landslide area, or only the scar of the landslide. What did the authors map? From the text, I cannot see if a landslide is presented by a (centroid) point, by the whole area, or something else. Please clarify how you mapped landslides.

Specific comments

Title

In my opinion, geoinformation properties is too vague. What about simply "effects of different land cover data on..."

Data

Effect of the different land cover data used – what I would be interested in, is also the extent of the influence of any land cover data at all. The difference between the

results of the two LUC data used suggests, that land cover does play a role – we do not fully know how significant it is (in this study area). I would be interested in seeing the difference between the two land cover data, and a susceptibility map without a land cover map. It would also be a sort of sensitivity analysis.

I would like to see the distribution of landslides (so, the points) on the data figure (Figure 2) as well (so, where are landslides located on a land cover map, soil map. . .)

Figure 4. I would like to see a different color map for the difference map. First of all, it would make sense, that there is a more logical center class. Currently, there are classes between 0.1 – 1 and -0.9 -0 (I assume, 0 is completely within this class). It would make more sense, to have a class -0.5 - +0.5 (or something similar).

Results

The authors compared the two maps visually, by map overlay, and by performing an overall accuracy and kappa coefficient. There is something I do not understand: what exactly is the overall accuracy? You compared the two maps, so this cannot be overall accuracy, is it maybe overall agreement? The same goes for commission and omission errors. These are not errors, but differences between two maps (so, two models). Also, I do not fully understand Table 4. From what I see in the table, most of the area is modelled as very high susceptibility in the study area – this however cannot be true. Or is the table presenting something else – maybe the susceptibility of the road network only? Please explain or modify the table.

Table 4 is one of the main results in my opinion, however, now you present it in % of total area. This is fine, but then you really need to replace the term accuracy with agreement, because 66.7% of accuracy (LSRN2/LSRN1) for the class high does not mean accuracy, but agreement.

Discussion

I already mentioned above what needs to be expanded in the discussion. Besides that,

I would like to see the following in the discussion: - Any recommendations based on the results? (in terms of using land cover data) - Comparison with other, similar studies, and what did they find out? - The influence of the method used (maybe information value results to fewer differences between using different land cover maps) - Discussion on other data, particularly landslide inventory (potentially missed landslides in forests, or type of mapping).

Technical corrections

In the abstract, the authors use the term "very good" when describing their models – please either replace it with a different term, or add justification for it being very good (e.g. both have an AUC over 0.9). Also, the AUC is not the only measure to address the model success, so I would refrain myself by using very good – you can state that the models have a high accuracy in terms of AUC or something similar.

The last sentence – landslide susceptibility maps are exactly what their name implies, maps providing information on how susceptible an area is to landslides. They are not maps, where landslides will probably occur. Please change this.

Generally, the level of English is high. Nevertheless, a spell check or rewriting of some parts of the manuscript is necessary: - The authors tend to use the word "the" too much in my opinion (the landslides, the total or partial, the landslide susceptibility assessment. . .). - Study area description, first sentence: simplify and write "We performed this study in Zezere. . ."). Also, what does "high slopes" mean? Steep slopes? Same goes for low slopes.

The authors use a lot of abbreviations – while some are presented in the main body, some are presented only in the abstract (e.g. LUC, COS, CLC). I recommend that you again define the abbreviations in the main text, when you use them for the first time.

---

## Referee Comment (RC2) · Anonymous Referee #2 · 7 Oct 2018

This paper analyses landslide susceptibility for an area in Portugal using standard input data and also conventional bivariate statistical analysis. From a methodological point of view, the paper doesn't provide new approaches or insights. The aim was to see what would be the effect of different land use/land cover maps on the overall prediction of landslides. For that two land use maps were used with a different level of detail. Although the authors acknowledge the importance of land use/land cover changes for the occurrence of landslides, they do not make an attempt to use a map of land cover changes as input for their analysis. While this could have been done with the use of multi-temporal satellite images, and also correlate this with the changes in landslides that occurred after these changes. Now the relationship between land use/land cover

remains vague throughout the paper. It is also not clear when the two land cover maps were made and how these relate to the landslides mapped from images of 2005. Parts of this study area have been affected seriously by forest fires in the past years, and this must have also resulted in higher landslide activity. Nothing of that is mentioned in the paper, and a multi-temporal analysis is also lacking. The relation between the two land use maps should also be presented more in detail: how do the classes overlap? And are differences caused by errors or by temporal changes? Are landslide more frequent in zones where the classification do not match? The relationship between the factor maps is considered as a bivariate relation only, whereas it is a multivariate problem. It matters to know what the slope steepness is in order to assess the importance of different land cover classes for landslide susceptibility. Landslide susceptibility maps are not validated using independent data sets that were not used for making the model. This is not how it should be done. The authors do not develop a specific method for landslide susceptibility along the road, but basically, overlay the susceptibility maps of the two landcover maps with the road network. The assessment of landslide susceptibility along road requires a different approach where engineered slopes and natural slopes are evaluated separately, and where homogeneous road section is outlined with the upslope areas that could influence it. The method presented here is too simple for practical use along roads. The paper does not address the issue of landslide runout, which in the case of roads might be one of the most important hazards: debris (flows) or rock falls from the upslope areas are likely to affect the road. Only addressing landslide initiation is not considered appropriate in such a case. The level of English is problematic, and the text needs to be thoroughly reviewed by an English editor. The paper also uses too many abbreviations which makes it very difficult to read. For example GI, MMU,LUC, COS, CLC, PFM, IV, Ai, Ri, LSM, LSRN. . . The paper refers to other publications of the authors which seem to have a substantial overlap with this manuscript. Some detailed comments: 1/23: locals should be locations 2/1: The landslides.. should be Landslides. The entire sentence should be rewritten 2/9: same 2/10: landslide occurrence 2/16-19: The entire sentence is not clear should be rewritten. I

would not use the abbreviation GI throughout the paper. Just mention factor maps. 2/23: of landslides 2/23- : you indicate the importance of land use dynamics but do not analyse it yourselves in this paper. 3/8-9: avoid GI . Improve the sentence 3/10-13: is this not the same topic as this paper? 3/16: improve description between brackets. 3/21-22: I don't think you achieved this goal 3/24: what does MMU mean? It is another abbreviation one has to remember. 4/6: high slope: steep slopes 4/10: rankers? 4/15: artificial land? 4/17-18: Improve the sentence 5/15- : there is no need to explain why you use slope steepness as a factor in landslide susceptibility assessment 6/5: how does it reflect moisture retention? What about the dynamic aspect of soil moisture? Table 1: include the date of production. 7/8: why such a coarse scale? 1:500000 for roads is too general. Figure 2: Are all these maps needed? Where is the landslide inventory map? 9/9 and table 2: round off values. What is the size frequency distribution? 10: is it needed to describe this method again? 11: this could actually be better illustrated with a small example, or otherwise referring to the source paper. 12/11-12: explain why this is done? Shouldn't this be based on the final score? Why 10 classes? What is the use of this for the end user of the susceptibility map? 13/15-16: if these are landslide scars then the landslides are not caused by it, but they result in bare areas. 14/15: success rate curves: validation should be done with independent data. What would be the result if you don't use any land-use map? 15/6-9: I don't understand what you are saying here. Explain it better. 15/11-14: not clear. Figure 6: the land use classes should be combined with slope . It is difficult to find out what the codes mean. There is not much description of it in the text. 16-17: I got lost in reading these pages with so many abbreviations and English language issues. Table 4: Not clear what the values indicate? Percentage of the area? Then the combinations are very strange: 86% in very high, and only 0.23 in very low? Figure 10 could be skipped.

---

## Author Comment (AC1) · 16 Nov 2018

Dear Reviewer 1,

the authors thank the comments. These are very relevant and well prepared and most of them were considered in the reviewing process. We believe that your contribution helped to improve the manuscript.

The manuscript addresses the effect of using land cover data of different spatial and thematic resolution in landslide susceptibility modeling, particularly for susceptibility zonation of a road network. The topic is interesting and significant. Land cover data is used in susceptibility mapping, often without questioning its quality and suitability for such an analysis. The authors provide us with an overview of what we might be missing in case we use unsuitable data (or data which is too coarse for example).

The authors first introduce us to landslides in general, and their effect on human lives, activities, infrastructure, etc. They proceed with describing the usefulness of landslide susceptibility assessments. Afterwards, we are introduced to the role of land cover and how the choice of geoinformation details of land cover is usually not studied – despite being a significant factor. The authors demonstrate their approach in a watershed in Portugal (with a detailed focus on three smaller areas within the watershed boundaries).

They use two different land use and land cover data to demonstrate the effect of using different data on landslide susceptibility: the Portuguese land cover map (COS), and the European Corine Land Cover (CLC). The landslide susceptibility mapping itself is straightforward and is based on acknowledged and commonly used methods (information value). Although, there are other approaches, where similar data could be used (logistic regression, weights of evidence), the authors chose this method, as it has been applied at a similar scale, also in Portugal. Generally, it is a nice study, however with some major flaws - most of them related to the fact that some topics were not addressed. This means, that my revision recommendation is mostly based on rewriting the main body of the text, adding additional clarifications, or expanding the discussion.

Aditional analyses are not needed.

First, while the authors investigated the role of land cover data on modeling landslide susceptibility, they did not compare different methods. I do not expect the authors to perform additional analyses with other modeling approaches – however, I would like them to discuss the method used a bit more extensively. For example, could other methods lead to larger (or smaller) differences between different land cover data?

Authors: We believe that the use of different landslide susceptibility methods culminated in different results, many studies highlight some differences in results obtained by different methods, specially between IV, logistic regression and weights of evidence. This fact is well developed scientifically, and the manuscript goal is not to compare results with different methods, but LUC with different properties. If the conditions are the same in the modeling process (predisposing factors, method, software, …), the differences in results can be derived from the change the LUC data, and its properties justify these differences. However, we introduced a reference about this point in discussion.

*"the differences observed in the landslide susceptibility models are a consequence of using different LUC data inputs (different properties), because the other predisposing factor maps are the same in two models."*

Second, I have seen many landslide susceptibility studies, where data with relatively coarse data has been used. Also here, the data on soil and lithology is on a (much) coarser resolution than other data (slope characteristics, and land cover). We can see, that while all other data has a fine and detailed pattern, the lithology and soil maps have clear boundaries, with relatively large mapping units. This is of course not the authors fault – this are probably still the most detailed soil and lithology maps available for the study area. Nevertheless, I would like to see a more detailed section in the discussion, reflecting on the discrepancy of such differences (e.g. scale and mapping unit) and the effects on susceptibility modeling. The authors already did this for the land cover maps, and wrote a few sentences in the discussion.

Authors: The data of soil and lithology used is the only free data available for the study area. Nevertheless, the soil data is incomplete in study area at scales ≥1:50 000 and this is also very expensive. The geological data is not available to the study area at the 1:50 000 scale (please check in http://www.lneg.pt/servicos/215/).

The predisposing maps (soil, lithology, slope, …) are statics in the landslide modelling process, except the LUC data, then the results change derived from the LUC data properties that integrated each landslide model. The IV can change if the scales of factor maps are different, is a fact, but we do not have the possibility to use more detailed geoinformation, specially information with high costs.

About this topic, a new sentence was introduced in discussion.

"*the data of soil and lithology was constrained to very generalized (1:1000000 and 1:50000 scales, respectively) and this factor can influence the IV results if more detailed data was considered in the modeling process. The performance of the landslide susceptibility mapping and assessment is controlled by the quality of the available data and it depends not only on the method.*"

Third, I have two comments regarding the landslide inventory (page 9, lines 5-10). To be clear, the authors mapped landslides using orthophotos and google earth themselves? And the authors went on the field themselves as well? Currently, it is not clear if they received the data that was developed by photointerpretation, or if they performed it themselves. Moreover, it seems that the most landslides are outside forested areas – I compared the two land cover maps with the landslide distribution map visually.

This is of course possible, as evidence shows that forests have a positive role on slope stability (e.g. due to roots). Nevertheless, it is also difficult to map landslides in forests, using photointerpretation only. This can have significant effects on the results.

For example, if we look at Figure 4, the areas where the differences between the two susceptibility maps are the lowest are indeed areas covered by forests (or seem to be, the authors did not provide additional information that would lead to other conclusions). Also, studies have shown that landslide unit definition have a significant effect on landslide susceptibility modeling. It makes a difference if a landslide is mapped as a point, as the whole landslide area, or only the scar of the landslide. What did the authors map? From the text, I cannot see if a landslide is presented by a (centroid) point, by the whole area, or something else. Please clarify how you mapped landslides.

Authors: The landslides were inventoried and validated only by authors. In fact, the photointerpretation of landslides in forest areas is very complex, and possibly some landslides cannot be inventoried or validated in the field because are covered by vegetation. Some landslides in burned areas were also considered.

The landslides are not represented with points, but polygons (areas) that represent the unstable area (scarp, body and toe), see the table 3 (Statistics description of the landslides inventory). Additional information was introduced in the Data section.

*"The landslide inventory was obtained by photointerpretation (orthophotos of the year 2005 and Google Earth images), a process supported by ancillary topographic data and further field work validation only performed in the sample areas (Fig. 1) due to the extension of the study area. A total of 128 landslides (predominantly shallow translational slides), with a total area of 74042 m2, was validated during field work in sample areas (49.4% of the total inventoried landslide cases). Among the landslides initially inventoried by photointerpretation in sample areas more than 90% of cases were confirmed. In these sample areas roads disruptions were also validated."*

Specific comments

Title

In my opinion, geoinformation properties is too vague. What about simply "effects of different land cover data on: : :"

Authors: We accept your suggestion and the title will be changed accordingly.

Data

Effect of the different land cover data used – what I would be interested in, is also the extent of the influence of any land cover data at all. The difference between the results of the two LUC data used suggests, that land cover does play a role – we do not fully know how significant it is (in this study area). I would be interested in seeing the difference between the two land cover data, and a susceptibility map without a land cover map. It would also be a sort of sensitivity analysis.

I would like to see the distribution of landslides (so, the points) on the data figure (Figure 2) as well (so, where are landslides located on a land cover map, soil map...)

Authors: We present a new table in supplementary data with the importance of each LUC type to landslide occurrences inventoried and also the distribution this LUC classes by slope classes, because this is also an important variable to landslides occurrence.

LUC data was tabulated (COS and CLC) and represented in table 2. The results show a distinction of the LUC types with principal differences between CLC Vs COS. The importance of this LUC variables is presented in table 4 and it is not necessary a new map to assess their importance in modelling. The construction of a new susceptibility landslide map without LUC data will be developed in a further study.

The landslides areas will be represented in Figures 2 and 5.

Figure 4. I would like to see a different color map for the difference map. First of all, it would make sense, that there is a more logical center class. Currently, there are classes between 0.1 – 1 and -0.9 -0 (I assume, 0 is completely within this class). It would make more sense, to have a class -0.5 - +0.5 (or something similar).

Authors: we acknowledge the reviewer comments. The colors and classes of this map were corrected in order to be represented with a more logical center and constant interval.

Results

The authors compared the two maps visually, by map overlay, and by performing an overall accuracy and kappa coefficient. There is something I do not understand: what exactly is the overall accuracy? You compared the two maps, so this cannot be overall accuracy, is it maybe overall agreement? The same goes for commission and omission errors. These are not errors, but differences between two maps (so, two models). Also, I do not fully understand Table 4. From what I see in the table, most of the area is modelled as very high susceptibility in the study area – this however cannot be true. Or is the table presenting something else – maybe the susceptibility of the road network only? Please explain or modify the table.

Table 4 is one of the main results in my opinion, however, now you present it in % of total area. This is fine, but then you really need to replace the term accuracy with agreement, because 66.7% of accuracy (LSRN2/LSRN1) for the class high does not mean accuracy, but agreement.

Authors: The overall "accuracy" will be changed to overall agreement, and "errors" by differences. The table present errors in the column headers due to a mistake in the copy process. The classes will be corrected. The class "very low" can never have 86.11% of total study area! Thank you for reporting us this error.

Discussion

I already mentioned above what needs to be expanded in the discussion. Besides that, I would like to see the following in the discussion: - Any recommendations based on the results? (in terms of using land cover data) - Comparison with other, similar studies, and what did they find out? - The influence of the method used (maybe information value results to fewer differences between using different land cover maps) - Discussion on other data, particularly landslide inventory (potentially missed landslides in forests, or type of mapping).

Authors: We will introduce some recommendations based on results, and discussion the problem of landslide inventory (potentially missed landslides in forests, or type of mapping).

"*More detailed LUC data (COS) allows better landslide susceptibility results, while LUC data is more generalized than CLC data, resulted in the IV reduction, not allowing identify some places where landslides occurred effectively.*"

"*The assignment of landslide susceptibility results to the road network allowed to identify the locations with the highest spatial probability to the landslide occurrence*"

Technical corrections

In the abstract, the authors use the term "very good" when describing their models –

please either replace it with a different term, or add justification for it being very good (e.g. both have an AUC over 0.9). Also, the AUC is not the only measure to address the model success, so I would refrain

myself by using very good – you can state that the models have a high accuracy in terms of AUC or something similar.

Authors: We agree with your comment and changes will be made accordingly.

The last sentence – landslide susceptibility maps are exactly what their name implies, maps providing information on how susceptible an area is to landslides. They are not maps, where landslides will probably occur. Please change this.

Authors: Thank you for the comment. Changes will be made accordingly.

Generally, the level of English is high. Nevertheless, a spell check or rewriting of some parts of the manuscript is necessary: - The authors tend to use the word "the" too much in my opinion (the landslides, the total or partial, the landslide susceptibility assessment: : :).

Authors: The English will be reviewed by English editor services.

- Study area description, first sentence: simplify and write "We performed this study in Zezere: : :).

Authors: OK.

Also, what does "high slopes" mean? Steep slopes?

Authors: high slopes will be changed by "steep slopes".

Same goes for low slopes.

Authors: low slopes will be changed by "gentle slopes"

The authors use a lot of abbreviations – while some are presented in the main body, some are presented only in the abstract (e.g. LUC, COS, CLC). I recommend that you again define the abbreviations in the main text, when you use them for the first time.

Authors: Some abbreviations will be decoded and eliminated (e.g. MMU, AUR, SRC, PFM). Other will be defined in the text when used for the first time.

---

## Author Comment (AC2) · 16 Nov 2018

Dear reviewer 2,

thanks for the comments. These are very relevant and well prepared and most of them were considered in the reviewing process. We believe that your contribution helped to improve the manuscript.

This paper analyses landslide susceptibility for an area in Portugal using standard input data and also conventional bivariate statistical analysis. From a methodological point of view, the paper doesn't provide new approaches or insights. The aim was to see what would be the effect of different land use/land cover maps on the overall prediction of landslides. For that two land use maps were used with a different level of detail. Although the authors acknowledge the importance of land use/land cover changes for the occurrence of landslides, they do not make an attempt to use a map of land cover changes as input for their analysis. While this could have been done with the use of multi-temporal satellite images, and also correlate this with the changes in landslides that occurred after these changes.

Now the relationship between land use/land cover remains vague throughout the paper. It is also not clear when the two land cover maps were made and how these relate to the landslides mapped from images of 2005. Parts of this study area have been affected seriously by forest fires in the past years, and this must have also resulted in higher landslide activity. Nothing of that is mentioned in the paper, and a multi-temporal analysis is also lacking.

Authors: the relationship between land use/land cover (LUC) is referred in introduction, and this case study, is evaluated its importance in landslide susceptibility zonation (now table 4). The LUCC was not evaluated in the present research, because the main goal of this work is the comparison between the landslide susceptibility results obtained with different LUC datasets (same predisposing factor, but with different properties). The guidelines of drawing up LUC maps are presented in the text, but we consider important to resume the properties of this LUC geoinformation to the reader in a Table.

The wildfires were evaluated by us in other research's (e.g. see Meneses *et al.*, 2018a), but the LUC maps (COS or CLC) do not represent the total burned areas, because if there is a potential to vegetation regeneration, the technical guidelines refer that the LUC type with this potential correspond to forest or scrubs, not the burned area observed in photointerpretation. By other side, this wildfires information does not interfere in the research goal, because burned areas were indirectly represented in the classes "Open spaces with little or no vegetation" and "Scrub and/or herbaceous vegetation associations". However, we also observed that burned areas do not match the principal location of the landslide inventory.

The landslide inventory was obtained by photointerpretation (orthophotos of the year 2005 and Google Earth images - 2004, 2005 and 2006), so these dates of information support the inventorying process selected according to LUC dates available (2006 and 2007). If these landslides were old slope movements, it would be more difficult to be identified through this information because of the regeneration of the vegetation.

The relation between the two land use maps should also be presented more in detail: how do the classes overlap? And are differences caused by errors or by temporal changes? Are landslide more frequent in zones where the classification do not match?

Authors: The relation between two land use maps was made and results are presented in Table 2. The main discrepancies were observed in forest areas and scrub and/or herbaceous vegetation associations, especially in central sector of watershed (surround of Cabril dam). By the landslides inventoried GIS analysis, we do not observed landslides in the areas with main discrepancies between COS and CLC (for the LUC types before mentioned). Some explanations will be made in the text.

Table 2. LUC data agreement (area ha) between CLC and COS classes.

| Data | COS | | | | | | | | | | | | |
|---|---|---|---|---|---|---|---|---|---|---|---|---|---|
| CLC | Urban fabric (UF) | Industrial, commercial and transport units (ICT) | Mine, dump and construction sites (MDC) | Artificial, non-agricultural vegetated areas (ANA) | Arable land (AL) | Permanent crops (PC) | Pastures (P) | Heterogeneous agricultural areas (HAA) | Forests (F) | Scrub and/or herbaceous vegetation associations (SHV) | Open spaces with little or no vegetation (OSV) | Inland waters (IW) | Total |
| UF | 3160.2 | 439.8 | 77.3 | 100.8 | 207.7 | 502.0 | 15.7 | 929.2 | 337.7 | 251.5 | 0.1 | 18.7 | 6040.7 |
| ICT | 134.1 | 650.4 | 83.0 | 9.5 | 33.4 | 27.4 | 9.0 | 62.5 | 130.8 | 207.7 | 0.3 | 8.1 | 1356.1 |
| MDC | 6.1 | 58.3 | 283.0 | 0 | 3.6 | 3.6 | 6.8 | 6.5 | 48.2 | 53.5 | 0.2 | 5.4 | 475.0 |
| ANA | 29.3 | 2.9 | 0 | 22.5 | 0 | 0 | 0 | 0 | 1.7 | 9.1 | 0 | 0 | 65.6 |
| AL | 245.3 | 171.7 | 25.0 | 12.2 | 9166.1 | 1304.4 | 2225.0 | 1317.1 | 1133.2 | 1435.9 | 51.0 | 190.7 | 17277.5 |
| PC | 1271.4 | 93.3 | 37.3 | 21.2 | 1357.9 | 7948.5 | 315.4 | 2930.0 | 2004.5 | 2300.2 | 7.9 | 38.1 | 18325.7 |
| P | 4.4 | 2.4 | 0 | 0 | 61.3 | 0.9 | 36.1 | 58.4 | 41.2 | 188.6 | 0 | 0 | 393.2 |
| HAA | 7791.6 | 736.5 | 271.4 | 73.7 | 11773.1 | 15553.2 | 2341.0 | 23762.4 | 16514.4 | 12935.5 | 143.3 | 243.9 | 92140.0 |
| F | 745.3 | 392.9 | 173.1 | 29.3 | 741.9 | 1715.5 | 238.1 | 4058.7 | 100486.5 | 26805.7 | 42.0 | 735.8 | 136164.8 |
| SHV | 826.5 | 510.0 | 259.3 | 38.0 | 1353.1 | 2543.2 | 958.3 | 5832.8 | 50509.8 | 149644.0 | 4052.8 | 846.7 | 217374.5 |
| OSV | 29.4 | 13.8 | 5.3 | 1.4 | 18.3 | 10.3 | 10.7 | 140.4 | 860.0 | 6367.1 | 4206.6 | 30.3 | 11693.7 |
| IW | 5.6 | 12.0 | 0 | 0.2 | 1.3 | 7.5 | 0 | 15.2 | 278.5 | 180.7 | 2.4 | 4589.5 | 5093.0 |
| Total | 14249.1 | 3084.1 | 1214.7 | 308.8 | 24717.7 | 29616.3 | 6156.0 | 39113.2 | 172346.6 | 200379.5 | 8506.6 | 6707.1 | 506399.7 |

The relationship between the factor maps is considered as a bivariate relation only, whereas it is a multivariate problem. It matters to know what the slope steepness is in order to assess the importance of different land cover classes for landslide susceptibility.

Authors: The importance of each class of explanatory variables to landslides occurrence was evaluated by conditional probabilities that integrated the Eq. 1. We also present new information about the slopes and LUC relation (supplementary data - tables) and more information about this point was introduced in the text.

"In general terms, slope angle increasing promotes the landslide occurrence and is a very good proxy of the shear stress (Zêzere et al., 2017). Slope instability is more frequent in higher slope angles of the Estrela Mountain and throughout Zêzere valley. Also, in these areas, convex slope curvature is predominantly related with slope instability. The slope aspect is important in the spatial distribution of the different LUC types of the study area (Fig. 2) and on slope instability, especially in northwest-facing slopes (more exposed to the rain and with higher humidity levels)."

Extract of supplementary data - Conditional and priori probabilities (CP and PP, respectively) of landslides occurrence in Zêzere watershed.

| PFM | Classe | Area watershed (%) | Landslides test area (%) | CP | PP | IV |
|---|---|---|---|---|---|---|
| Slope angle | 0 - 5 | 28,17 | 1,69 | 0,000098 | 0,001652844 | -2,824 |
| | 06-10 | 18,22 | 2,07 | 0,000206 | 0,001652844 | -2,083 |
| | 11v-15 | 17,93 | 5,73 | 0,000617 | 0,001652844 | -0,986 |
| | 16 - 20 | 15,30 | 12,97 | 0,001433 | 0,001652844 | -0,143 |
| | 21 - 25 | 10,94 | 17,48 | 0,002798 | 0,001652844 | 0,526 |
| | 26 - 30 | 6,03 | 17,95 | 0,005499 | 0,001652844 | 1,202 |
| | 31 - 35 | 2,47 | 17,76 | 0,01118 | 0,001652844 | 1,912 |
| | 36- 40 | 0,73 | 10,71 | 0,020153 | 0,001652844 | 2,501 |
| | 41- 45 | 0,16 | 10,90 | 0,090941 | 0,001652844 | 4,008 |
| | 46 - 50 | 0,03 | 2,54 | 0,159236 | 0,001652844 | 4,568 |
| | 51 -55 | 0,01 | 0,19 | 0,042391 | 0,001652844 | 3,244 |
| | > 55 | 0,01 | 0 | 0 | 0,001652844 | -2,824 |

Landslide susceptibility maps are not validated using independent data sets that were not used for making the model. This is not how it should be done.

Authors: we acknowledge the reviewer comment. The research was reformulated, and independent dataset were used in order to perform an independent validation of the landslide susceptibility. The

landslide inventory was randomly divided in two subsets (Fig. 1): the landslide training group and the landslide test group. The first group integrated the modelling an the second the validation process. More explanations about this procedure was introduced in the text.

*"The landslide inventory was obtained by photointerpretation (orthophotos of the year 2005 and Google Earth images), a process supported by ancillary topographic data and further field work validation only performed in the sample areas (Fig. 1) due to the extension of the study area. A total of 128 landslides (predominantly shallow translational slides), with a total area of 74042 m2, was validated during field work in sample areas (49.4% of the total inventoried landslide cases). Among the landslides initially inventoried by photointerpretation in sample areas more than 90% of cases were confirmed. In these sample areas roads disruptions were also validated.*

*For complete Zêzere watershed 259 landslides have been identified, predominantly of shallow type. On the total, 32 landslides affected directly the road network (total or partial blockages by the material and 7 cases with partial loss of infrastructure). The landslide inventory was randomly divided in two subsets (Fig. 1) (Chung and Fabbri, 2003): the landslide training group and the landslide test group (81.5% and 18.5% of the total landslide affected area, respectively). The statistical description of each landslide group is presented in Table 3."*

**Table 3.** *Statistics description of the training group and test group landslide inventories.*

|  | Training group | | Test group | | Total inventory |
|---|---|---|---|---|---|
|  | Non affected roads | Affected roads | Non affected roads | Affected roads |  |
| **Total landslides** | 185 | 26 | 42 | 6 | 259 |
| **Total area (m²)** | 44604 | 369404 | 10444 | 12089 | 104077 |
| **Minimum (m²)** | 134 | 7 | 18 | 82 | 7 |
| **Maximum (m²)** | 27364 | 12507 | 1911 | 5881 | 12507 |
| **Mean (m²)** | 2414 | 1421 | 249 | 2015 | 402 |
| **Std. deviation (m²)** | 3284 | 2647 | 304 | 2627 | 1069 |

The authors do not develop a specific method for landslide susceptibility along the road, but basically, overlay the susceptibility maps of the two landcover maps with the road network.

The assessment of landslide susceptibility along road requires a different approach where engineered slopes and natural slopes are evaluated separately, and where homogeneous road section is outlined with the upslope areas that could influence it. The method presented here is too simple for practical use along roads.

Authors: We don't simply overlay the susceptibility maps of the two landcover maps! The LUC maps integrated only the susceptibility modeling and the results was integrated in the road network (different datasets), allowing these results the representation of the landslide susceptibility obtained in context more widely, not point by point and assessed in isolation by each segment of roads.

The paper does not address the issue of landslide runout, which in the case of roads might be one of the most important hazards: debris (flows) or rock falls from the upslope areas are likely to affect the road. Only addressing land- slide initiation is not considered appropriate in such a case.

Authors: We present some examples of landslides validated in study area (Fig. 1). In the landslide susceptibility model only landslides were considered (predominantly shallow translational slides of small area and length).

The level of English is problematic, and the text needs to be thoroughly reviewed by an English editor.

Authors: The text will be reviewed by an English editor.

The paper also uses too many abbreviations which makes it very difficult to read. For ex- ample GI, MMU, LUC, COS, CLC, PFM, IV, Ai, Ri, LSM, LSRN. . .

Authors: Some abbreviations were decoded and eliminated (e.g. MMU, AUR, SRC, PFM). Other were defined in the text when used for the first time.

The paper refers to other publications of the authors which seem to have a substantial overlap with this manuscript.

Authors: The manuscript is an original research and the other publications do not focus on the same goals of this work. The study area is very important in Portugal, because have important supply water bodies and this fact justify the many publications of authors in this watershed, although not overlapping in the goals and results of the presented work. None of the published work addresses the issue of landslides or uses the presented methodology.

Some detailed comments:

- 1/23: locals should be locations Authors: OK
- 2/1: The landslides.. should be Landslides. The entire sentence should be rewritten Authors: OK
- 2/9: same Authors: ok
- 2/10: landslide occurrence Authors: OK
- 2/16-19: The entire sentence is not clear should be rewritten. I would not use the abbreviation GI throughout the paper. Just mention factor maps. Authors: OK
- 2/23: of landslides Authors: ok
- 2/23- : you indicate the importance of land use dynamics but do not analyse it yourselves in this paper. Authors: Yes, this is an introduction, and the main goal isn't the dynamics of LUC assessment.
- 3/8-9: avoid GI. Improve the sentence Authors: OK
- 3/10-13: is this not the same topic as this paper?
  Authors: The references present two different researches: 1 – LUCC in Portugal: multi-scale and multi-temporal differences obtained by LUC of different years; 2 – The paper assess the effects LUC geoinformation raster generalization in the analysis of LUCC in Portugal using different LUC datasets. None of the published work addresses the issue of landslides or uses the presented methodology.
- 3/16: improve description between brackets. Authors: OK
- 3/21-22: I don't think you achieved this goal
  Authors: This goal was achieved, because we present different results about the landslide susceptibility zonation of road network derived of integration LUC GI with different properties in the models, and we explain why in manuscript.
- 3/24: what does MMU mean? It is another abbreviation one has to remember.
  Authors: This abbreviation was decodified in the text.
- 4/6: high slope: steep slopes Authors: OK
- 4/10: rankers? Authors: OK
- 4/15: artificial land? Authors: OK
- 4/17-18: Improve the sentence Authors: OK

- 5/15- : there is no need to explain why you use slope steepness as a factor in landslide susceptibility assessment
  Authors: this is a complementary information for the readers and explain part of landslides, for example in Estrela mountain.
- Table 1: include the date of production. Authors: OK
- 7/8: why such a coarse scale? 1:500000 for roads is too general.
  Authors: This is the data validated available to research (free data). Other information is available, but with considerable costs. However, at the scale of research, the GI used represent the main road network and serve the purposes of this research.
- Figure 2: Are all these maps needed? Where is the landslide inventory map?
  this is the predisposing factor maps used in many researches of landslide susceptibility in Portugal and we explain why in the introduction and, also, in characterization of study area. The landslide inventory is represented in Figure 1 and the characteristics in Table 3.
- 9/9 and table 2: round off values. Authors: OK
- What is the size frequency distribution?
  Authors: the frequency is represented in fig 3.
- 10: is it needed to describe this method again? Authors: recast text.
- 12/11-12: explain why this is done? Shouldn't this be based on the final score? Why 10 classes? What is the use of this for the end user of the susceptibility map?
  Authors: Landslide susceptibility maps were built and classified in 10 classes (deciles) because allow the visual comparison between results of different models. We performed some tests to represent IV by classes and the deciles method present good results allowing the comparison above mentioned.
- 13/15-16: if these are landslide scars then the landslides are not caused by it, but they result in bare areas.
  Authors: this is an assumption generalized, but the forest also includes landslides. Please, see the extract table with the landslide area by LUC type in supplementary data.

*Extract of supplementary data.*

| PFM | Classe | Area watershed (%) | Landslides test area (%) |
|---|---|---|---|
| COS | Urban fabric | 2,81 | 0,72 |
| | Industrial, commercial and transport units | 0,61 | 0 |
| | Mine, dump and construction sites | 0,24 | 0 |
| | Artificial, non-agricultural vegetated areas | 0,06 | 0 |
| | Arable land | 4,88 | 0 |
| | Permanent crops | 5,85 | 0,84 |
| | Pastures | 1,22 | 0 |
| | Heterogeneous agricultural areas | 7,72 | 0,96 |
| | Forests | 34,03 | 14,34 |
| | Scrub and/or herbaceous vegetation associations | 39,57 | 81,96 |
| | Open spaces with little or no vegetation | 1,68 | 1,19 |
| | Inland waters | 1,32 | 0 |
| CLC | Urban fabric | 1,19 | 0 |
| | Industrial, commercial and transport units | 0,27 | 0 |
| | Mine, dump and construction sites | 0,09 | 0 |
| | Artificial, non-agricultural vegetated areas | 0,01 | 0 |
| | Arable land | 3,41 | 0 |
| | Permanent crops | 3,62 | 0 |
| | Pastures | 0,08 | 0 |
| | Heterogeneous agricultural areas | 18,20 | 1,91 |
| | Forests | 26,89 | 22,10 |
| | Scrub and/or herbaceous vegetation associations | 42,93 | 71,57 |
| | Open spaces with little or no vegetation | 2,31 | 4,42 |
| | Inland waters | 1,01 | 0 |

- 14/15: success rate curves: validation should be done with independent data. What would be the result if you don't use any land-use map?
  Authors: the landslide susceptibility validation was made with landslide test group (1), and we also assessed the performance of models with a part of the landslide inventory (training group), and now

prediction and success rate curves will be presented. The partition inventory increases the performance of models (see AUC), comparatively to the results presented in first version.

By other side, the importance of each variable in model's is presented in table 4 and LUC is important in the analysis, i.e., the determined classes of this predisposition factor are relevant because they contain landslide area.

- 15/6-9: I don't understand what you are saying here. Explain it better. We will make an effort to clarify this idea in the text.
- 15/11-14: not clear. Authors: ok
- Figure 6: the land use classes should be combined with slope. It is difficult to find out what the codes mean. There is not much description of it in the text.
  Authors: we introduce the decoding after figure. 16-17: I got lost in reading these pages with so many abbreviations and English language issues.
  Authors: the abbreviations were reduced.
- Table 4: Not clear what the values indicate? Percentage of the area? Then the combinations are very strange: 86% in very high, and only 0.23 in very low?
  Authors: table 4 (now table 5) represent the area (%) of watershed area by each susceptibility class, and when tabulation is performed between LSRN1 and LSRN2 the coincident area between the same classes and the area distributed by other classes is obtained. In final, it is possible to represent the agreement value between two maps.
- Figure 10 could be skipped. Authors: Ok

---

## Author Response (AR2)

**Dear Editor Thomas Glade,**

We thank the revision and we send point-by-point reply to the reviewer's comments and a marked–up manuscript version showing all the changes made in the manuscript.

Best regards,
Bruno Meneses
Susana Pereira
Eusébio Reis

Comments to Reviewer,
The authors thank the comments. These are very relevant and well prepared and most of them were considered in the reviewing process (second round). We believe that your contribution helped to improve the manuscript.

The authors improved the manuscript. I do still have issues with the discussions section, which is not written in the best way.
Authors: We thank the reviewer comments to improve the discussion section.
First, you should not start the discussion with "According to this research..." but with a more general statement on the importance of such work. Particularly, why does such a study matter at all? What is the innovative here, that the readers now
20 know and did not know before?
Authors: We acknowledge the reviewer comment and we have made an effort to start the discussion highlighting the importance of this study.
"In landslide hazard and risk assessment, the LUC data integrate the controlling factors group and, in many evaluations, is pointed by another factor input to the model. Usually LUC data is used as a landslide conditioning factor which, in some cases,
25 is scarce, generalized and low detailed. For example, Eeckhaut and Hervás (2012) verified that in the different locations of Europe the CLC is widely used for landslide assessment, because is the only LUC data available. Remote sensing and satellite images contributed to LUC data acquisition for landslide susceptibility assessment in different times (Guzzetti et al., 2012) and territories, and minimize some problems of scarcity and detail (thematic and resolution). LUC is an important conditioning factor in landslide susceptibility (Pisano et al., 2017), and the high accountability index results prove this fact (Table 4).
30 There are several studies about the influence of land use cover changes on landslide susceptibility (e.g., Karsli et al., 2009; Mugagga et al., 2012; Promper et al., 2014; Reichenbach et al., 2014), and differences between the susceptibility zonation

were obtained with the LUC maps of different dates, although to the best of our knowledge there are no approaches that analyze the influence of LUC data properties in the same or approximate date on the landslide susceptibility, especially in small areas (e.g., road networks).

When different LUC datasets are available, the choice for the LUC dataset used in the landslide susceptibility assessment is
5  not always clearly justified, and the results may vary according to LUC data properties selected. For Portuguese territory different LUC datasets (with different properties) are available, but the use of each dataset can generate different conclusions, for example, different land use and land cover changes in the same period were observed by Meneses et al. (2018c). "

This study is a contribution to understand:
10  1 – The importance of LUC as a landslide predisposing factor in this study area, as it was shown with the accountability results, that explains how different classes of predisposition factors are relevant in the landslide analysis because they contain the landslide area (Table 4).

2 – The use of different LUC datasets with different properties generate different landslide susceptibility results in spatial terms in the Zêzere watershed.
15  3 - We have shown that when more detailed LUC data (COS) are used, the landslide susceptibility shows a higher prediction capacity between the landslide susceptibility model and the landslide inventory.

Then, point out the obvious in susceptibility analysis - that the LUC data are often taken for granted, as just another input to the model. Can you say, based on your results, that you have shown how important LUC data are? Then you can provide a few
20  examples, where others tried to solve this, or where they failed.

Authors: We have provided some references of previous works that have discussed the use of the LUC data in the landslide susceptibility analysis at different scales. The important of LUC data was observed by the accountability index, in this case is very high, i.e., LUC is one of the most important variables in landslide susceptibility assessment (was evidenced in the manuscript). A problem pointed by different authors is the scarce LUC data (in time, but also in determined territories), and
25  many studies are performed using LUC data obtained by satellite images. However, this is not the main goal of this research, i.e., comparing landslide susceptibility in different times derived of LUCC. Now, the readers understand the importance of LUC data properties in this type of research (for study area), i.e., LUC data with different properties (but same date) generate different conclusions, and this is the most important contribution of this research, that is better evidenced in the manuscript.

30  You should compare your study with other, similar studies, and emphasize the issues you have solved (or new issues you have discovered). Then, you can discuss the specific parts of your analysis (the current discussion does only that - tries to explain individual parts of your research, but does not put it into context of the state of the art research).

Authors: There are several studies about the influence of land use cover changes on landslide susceptibility (e.g., Karsli et al., 2009; Mugagga et al., 2012; Promper et al., 2014; Reichenbach et al., 2014), although to the best of our knowledge there are

no approaches that analyze the influence of LUC data properties in the same or approximate date on the landslide susceptibility, in small areas (e.g., road networks). This fact was mentioned now in the manuscript.

Finally, I would like to see how this can influence future research, or decision making (that often relies on studies, where LUC data are taken for granted).

Authors: We have introduced a new paragraph in the discussion section to explain how this work can influence future research or decision making.

[revised manuscript text omitted]

---

## Author Response (AR3)

**Reply to editor:**

Dear editor,

We have doubts which final reviewer comments we should address from the reviewer report. So, we have
5  improved the answer to the three final sentences of the reviewer report of 17 December and we have no
further comments for this reviewer. We hope this is your request.

Also, we have checked typos, missing co-authors and their affiliations, terminology, updates of data in
tables, and updates of variables in equations. All new changes were signed in this new version of the
manuscript.

10 We request that the review process as swiftly as possible, because this manuscript has already been
submitted in 2017.

Best regards.

**Reply to reviewer:**

Reviewer comment: "You should compare your study with other, similar studies, and emphasize the
issues you have solved (or new issues you have discovered). Then, you can discuss the specific parts of
20 your analysis (the current discussion does only that - tries to explain individual parts of your research, but
does not put it into context of the state of the art research). Finally, I would like to see how this can
influence future research, or decision making (that often relies on studies, where LUC data are taken for
granted)."

25 Authors: We have made an effort to improve the discussion section and include the reviewer comments.

There are several studies about the influence of land use cover changes on landslide susceptibility (e.g.,
Karsli et al., 2009; Mugagga et al., 2012; Promper et al., 2014; Reichenbach et al., 2014), although to the
best of our knowledge there are no approaches that analyze the influence of different LUC datasets with
30 different properties (date and base maps used on the production, spatial resolution, scale, minimum
mapping unit, or others) on the landslide susceptibility results. When the landslide predisposing factors
are collected, the LUC dataset must be selected according to its abovementioned properties, and not only
on the basis of its availability and free of charge conditions. We believe that these questions are now clear
on the discussion section and further references were included to support the discussion.

[revised manuscript text omitted]